# MLL oncoprotein levels influence leukemia lineage identities

Derek H. Janssens [1,2], Melodie Duran [1], Dominik J. Otto [1,3], Weifang Wu[1], Yiling Xu[1,4], Danielle Kirkey[5,6], Charles G. Mullighan [7,8], Joanna S. Yi[9], Soheil Meshinchi[5,6], Jay F. Sarthy [10], Kami Ahmad [1] ✉ & Steven Henikoff [1,4] ✉

Chromosomal translocations involving the *mixed-lineage leukemia (MLL)* locus generate potent oncogenic fusion proteins (oncoproteins) that disrupt regulation of developmental gene expression. By profiling the oncoprotein-target sites of 36 broadly representative *MLL*-rearranged leukemia samples, including three samples that underwent a lymphoid-to-myeloid lineage-switching event in response to therapy, we find the genomic enrichment of the oncoprotein is highly variable between samples and subject to dynamic regulation. At high levels of expression, the oncoproteins preferentially activate either an acute lymphoblastic leukemia (ALL) program, enriched for pro-B-cell genes, or an acute myeloid leukemia (AML) program, enriched for hematopoietic-stem-cell genes. The fusion-partner-specific-binding patterns over these gene sets are highly correlated with the prevalence of each mutation in ALL versus AML. In lineage-switching samples the oncoprotein levels are reduced and the oncoproteins preferentially activate granulocyte-monocyte progenitor (GMP) genes. In a sample that lineage switched during treatment with the menin inhibitor revumenib, the oncoprotein and menin are reduced to undetectable levels, but ENL, a transcriptional cofactor of the oncoprotein, persists on numerous oncoprotein-target loci, including genes in the GMP-like lineage-switching program. We propose MLL oncoproteins promote lineage-switching events through dynamic chromatin binding at lineage-specific target genes, and may support resistance to menin inhibitors through similar changes in chromatin occupancy.

*MLL*-rearranged (*MLL*r) leukemias are defined by chromosomal translocations that produce potent in-frame fusion oncoproteins that associate with chromatin and cause upregulation of target gene expression[1,2]. *MLL*r leukemias are highly aggressive and traditionally have been associated with a poor prognosis[3,4]. Advances in targeted immuno-therapies and pharmacological inhibitors have dramatically improved the treatment options for *MLL*r leukemias[5,6]. Despite sharing genetically related oncogenic

[1]Basic Sciences Division, Fred Hutchinson Cancer Center, Seattle, WA, USA. [2]Department of Epigenetics, Van Andel Institute, Grand Rapids, MI, USA. [3]Translational Data Science IRC, Fred Hutchinson Cancer Center, Seattle, WA, USA. [4]Howard Hughes Medical Institute, Chevy Chase, MD, USA. [5]Translational Science and Therapeutics Division, Fred Hutchinson Cancer Center, Seattle, WA, USA. [6]Division of Hematology and Oncology, Seattle Children's Hospital, Seattle, WA, USA. [7]Department of Pathology, St. Jude Children's Research Hospital, Memphis, TN, USA. [8]Center of Excellence for Leukemia Studies, St. Jude Children's Research Hospital, Memphis, TN, USA. [9]Pediatric Hematology and Oncology, Baylor College of Medicine and Texas Children's Hospital, Houston, TX, USA. [10]Seattle Children's Research Institute, Seattle, WA, USA. ✉e-mail: kahmad@fredhutch.org; steveh@fredhutch.org

lesions, *MLL*r leukemias show highly heterogeneous responses to therapeutic intervention[5,7,8].

The acquisition of secondary mutations and clonal selection are common genetic mechanisms that lead to therapeutic resistance, however, *MLL*r leukemias are characterized by an unusually low mutational burden[9], and may also evade targeted therapies through epigenetic mechanisms that are not well understood. In response to therapeutic pressures, *MLL*r B-cell acute lymphoblastic leukemia (B-ALL) occasionally switches to an acute myeloid leukemia (AML) identity and rapidly relapses[10–13]. Several pharmacological agents have been developed that target the interaction of the MLL oncoproteins with menin, a chromatin scaffold protein, and cause dissociation of oncoprotein complexes from chromatin[14,15]. In phase I clinical trials of the menin inhibitor revumenib, reduced oncoprotein target gene expression was correlated with disease remission[5]. Mutations in *MEN1* (encoding menin) that block the interaction of menin with revumenib act as a genetic mechanism of resistance[16]. In addition, genetic mutations, alternative splicing and altered expression of chromatin regulatory proteins such as CHD4 have been identified in *MLL*r AML samples after lineage switching[10], and mutations in components of a MLL related complex, KMT2C/D-UTX, can lead to resistance to menin inhibitors in cell line and animal models[17]. This indicates that concerted alterations in gene expression can also lead to therapeutic resistance and relapses of *MLL*r leukemias.

Single-cell RNA sequencing and chromatin profiling studies have identified considerable intra-tumoral heterogeneity in *MLL*r leukemia samples[8,18], but why *MLL*r leukemias, as opposed to other genetic subtypes of acute leukemia, display an unusual propensity for lineage switching remains unclear. Previously, we developed a high-throughput method for mapping the genome-wide binding sites of MLL oncoproteins in patient samples and found the oncoprotein-target genes themselves are dynamically regulated[19], suggesting the oncoproteins may contribute to the lineage plasticity and epigenetic resistance to targeted therapies.

Here, by directly profiling the MLL oncoproteins in primary patient samples, we demonstrate that the oncoprotein fusion partner and its expression levels influence genome-wide binding site selection. We find that both the oncoprotein expression and target binding are dynamic and promote lineage switching by transitioning from a B-ALL-instructive role to activating genes that are normally expressed in granulocyte-monocyte progenitor (GMP) cells. We characterize a lineage-switching event that occurred during treatment with a menin inhibitor, and find that in the AML patient sample the oncoprotein cofactor ENL remains bound to a subset of oncoprotein-target genes including the GMP-like program, despite the on-target dissociation of the oncoprotein and menin from chromatin. In addition, we find this same GMP-like program is upregulated in a previously characterized menin-inhibitor-resistant AML, suggesting the persistent activation of this GMP-like program may be a recurring mechanism of epigenetic resistance that allows *MLL*r leukemias to evade targeted therapies.

## Results

### Heterogeneous MLL oncoprotein levels

We used AutoCUT&RUN[20] to profile MLL-oncoprotein binding sites in 36 *MLL*r leukemia samples, including 4 cells lines, 14 infant leukemias and 18 leukemias from pediatric or adult patients that are representative of the major lineage subtypes and the most common oncoproteins found in *MLL*r leukemias (Fig. 1a, Supplementary Fig. 1a–c, Supplementary Data 1). To identify sites that are specifically bound by the oncoprotein, we performed AutoCUT&RUN assays using antibodies targeting the N-terminal domain of MLL, which is present in the oncoprotein as well as the remaining wild-type copy of MLL, and antibodies targeting the C-terminal domain of MLL, which is specific to the wild-type protein (Fig. 1b). Then, we called oncoprotein target sites according to the ratio of the MLL N-terminal signal over the C-terminal

signal and determined the statistical significance of this ratio by considering the number of reads that fell in each interval (Fig. 1c). We set a threshold that produced 3 false-positive sites in the CD34+, oncoprotein-negative control sample, and identified 156 oncoprotein target sites in the SEM cell line (Fig. 1c). By applying this threshold to all 36 *MLL*r leukemia samples, we identified 1692 oncoprotein-binding sites and 1092 direct oncoprotein-target genes (Supplementary Data 2).

To characterize the degree of inter-tumoral heterogeneity in the oncoprotein-binding sites amongst *MLL*r leukemias, we assigned an "oncoprotein score" to all 1692 intervals that incorporate the MLL N over C terminal signal as well as the statistical significance of this value. We performed dimensionality reduction using principal component analysis and found the first four components captured greater than 50% of the variance in oncoprotein scores (Supplementary Fig. 2a). When organized in UMAP space according to the first four components, the *MLL*r leukemia samples grouped into four clusters (Fig. 1d, Supplementary Fig. 2b). This analysis indicates the lineage identity and translocation-partner genes both contribute to the genome-wide oncoprotein scores. For example, Cluster 2 is comprised exclusively of *MLL*r AML samples, and Clusters 3 and 4 are almost all ALLs (Fig. 1d). Cluster 4 is unique in that it contains only leukemias bearing the *MLL::AF4* translocation (Fig. 1e). The *AF4* locus is the most common *MLL* translocation partner, and this rearrangement is found almost exclusively in B-ALLs at diagnosis[21]. Our results show the MLL oncoproteins bind to distinct sites in AMLs versus ALLs, and in a subset of the ALL samples, the MLL::AF4 oncoproteins occupy chromatin at regions that distinguish them from other *MLL*r leukemias.

The three B-ALL samples that went on to lineage-switch are interspersed amongst the other B-ALL samples we profiled. Cluster 4 includes a B-ALL sample and the paired AML relapse sample (Fig. 1d, labeled α). Cluster 3 contains two B-ALLs that went on to lineage switch. The first bears a *MLL::ENL* rearrangement and lineage switched during treatment with a menin inhibitor, and the paired AML sample is in Cluster 1 (Fig. 1d, labeled β). The last lineage-switching sample we profiled bears a *MLL::EPS15* rearrangement (Fig. 1d, labeled γ). For this patient, a post-lineage-switching sample was not available, but we were able to profile a second sample collected one week prior to lineage switching that also grouped in Cluster 3. We conclude the genome-wide localization of the oncoprotein does not obviously distinguish the B-ALL samples that went on to lineage switch from the other B-ALL samples we profiled.

In addition to the different lineage identities and fusion partner genes, we detect significant heterogeneity in the oncoprotein levels within our collection of samples. For example, Cluster 1 includes the CD34+ oncoprotein-negative control sample as well as 8 AML samples, 2 infant ALLs, 1 mixed phenotype acute leukemia (MPAL), and an AML that resulted from a lineage-switching event that occurred during treatment with the menin inhibitor. The oncoprotein signal is barely detectable in these Cluster 1 samples (Fig. 1f). The average oncoprotein scores of Clusters 2 and 3 are significantly higher than those found in Cluster 1, and the oncoprotein scores are highest for samples in Cluster 4 (Fig. 1f,g). By performing oncogene specific qPCR on samples that share the same minimal *MLL::AF4*, *MLL::ENL* and *MLL::AFDN* exon junctions, we find the differences in oncoprotein scores we observed using CUT&RUN generally reflect differences in the expression levels of the oncogene (Fig. 1h, Supplementary Fig. 2c–e). Only two samples that both bear the *MLL::AFDN* rearrangement did not fit this trend (marked by a star and asterisk in Fig. 1h). The ML-2 cell line (star in Fig. 1h) lacks the wild-type copy of *MLL*, suggesting wild-type MLL may be required for efficient oncoprotein loading. The wild-type MLL C-terminal signal was detectable in the second sample (asterisk in Fig. 1h), but it is possible the oncoprotein loading efficiency is reduced in this sample through an alternative mechanism. We conclude the average oncoprotein score provides a semi-quantitative metric that is

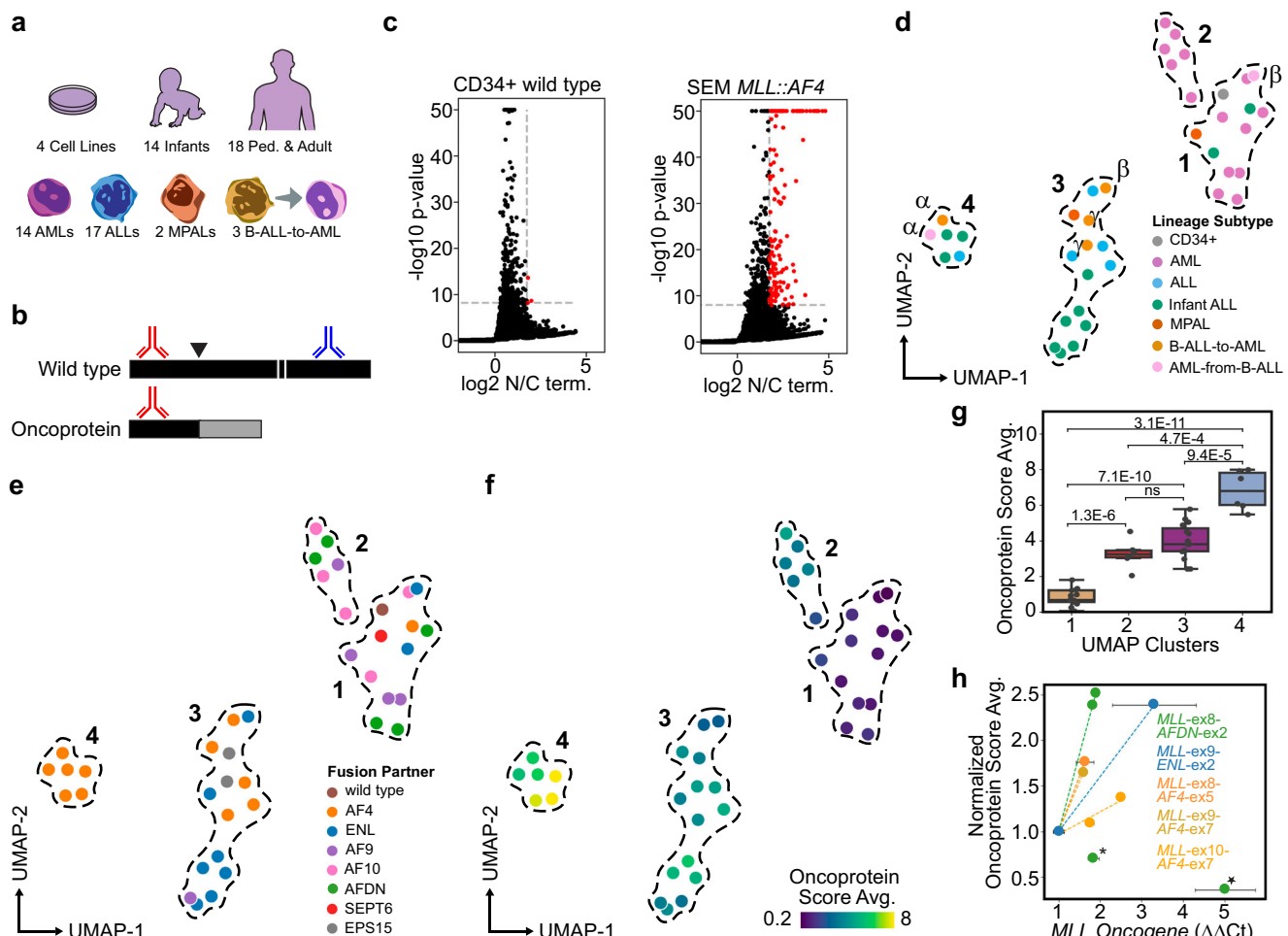

**Fig. 1 | Genomic enrichment of MLL oncoproteins is highly heterogeneous. a** We performed AutoCUT&RUN on a collection of *MLL*r cell lines and patient samples including a range of patient ages and lineage subtypes. **b** Comparison of CUT&RUN profiles using antibodies targeting the MLL N-terminal (red) and C-terminal (blue) regions can discriminate the oncoprotein-binding sites from wild-type MLL binding sites. **c** When the same magnitude and significance thresholds are applied to the MLL N versus C terminal profiles, we identify many more oncoprotein-target sites (red dots) in the SEM *MLL*r cell line than the CD34+ wild type control sample. *p* values were computed from the mean and standard deviation of Monte Carlo sampling of the N/C scores (*n* = 1000 samples) of each genomic interval; *p* values were corrected for multiple hypothesis testing using the Benjamini/Hochberg method. **d** Principal component analysis and UMAP embedding splits the *MLL*r samples into four clusters that are outlined with dotted lines; samples are colored according to the lineage subtype; patient-matched lineage-switching samples are indicated by the Greek letters α, β, γ. **e** The UMAP embedding from (**d**) but colored according to the MLL-oncoprotein-fusion partner. **f** Same as (**d**) colored according to the average oncoprotein scores (N/C terminal magnitude x significance). **g** The UMAP clusters capture differences in the global oncoprotein levels. *p* values were computed using a two-tailed independent samples t-test; Cluster 1 *n* = 13 samples, Cluster 2 *n* = 6 samples, Cluster 3 *n* = 15 samples, Cluster 4 *n* = 5 samples; boxplot center lines = median, box limits = first and third quartiles, whiskers = 1.5 times the interquartile range (IQR). **h** The average oncoprotein scores scale with the relative oncogene expression. Samples are grouped and colored according to the minimal *MLL*-fusion-partner exon junctions. Two *MLL::AFDN* samples were identified as outliers: ML-2 is indicated by a star and A70498 by an asterisk. Dotted lines indicate the regression, error bars indicate the standard deviation of the mean of three qPCR biological replicates. Source data are provided as a Source Data file.

indicative of differences in the oncogene expression levels between samples. Among leukemias bearing the most common *MLL* translocations, the gene expression levels of the oncoprotein are highly heterogeneous.

MLL oncoproteins are thought to promote transcriptional activation of their target genes[1,2]. To test this model, we examined whether differences in oncoprotein levels at specific target loci are correlated with target gene expression. *MBNL1* is the most frequently called oncoprotein target gene across the collection of leukemias we profiled (24/36 samples, Supplementary Data 2). We detected differences in oncoprotein binding on the *MBNL1* locus that are representative of the average oncoprotein scores in samples from each of the four UMAP clusters (Fig. 2a,b). In the *MLL::AF4* and *MLL::ENL* samples we profiled by qPCR, the *MBNL1* oncoprotein scores show a strong positive correlation with the relative expression levels of *MBNL1* (Fig. 2c). A

positive but less pronounced correlation is observed in the *MLL::AFDN* samples (Fig. 2c). *MEIS1, JMJD1C* and *MEF2C* are identified as oncoprotein target genes in 21/36, 22/36, and 21/36 of the samples, respectively (Supplementary Data 2). We observe a strong positive correlation between *MEIS1* oncoprotein scores and *MEIS1* gene expression, similar to the correlation seen with *MBNL1* (Supplementary Fig. 3a–c). However, MLL::AF4 and MLL::AFDN oncoprotein scores do not show a strong correlation with the expression of *JMJD1C* or *MEF2C* (Supplementary Fig. 3d–i). Our results are consistent with the model that MLL oncoproteins act as transcriptional activators, with some genes, like *MBNL1* and *MEIS1* being more sensitive to the MLL oncoprotein dosage than others like *JMJD1C* and *MEF2C*, which may be more susceptible to regulation by other factors.

Previously, we found the oncoprotein cofactors DOT1L and ENL are strongly enriched at MLL-oncoprotein target sites[19]. To test our

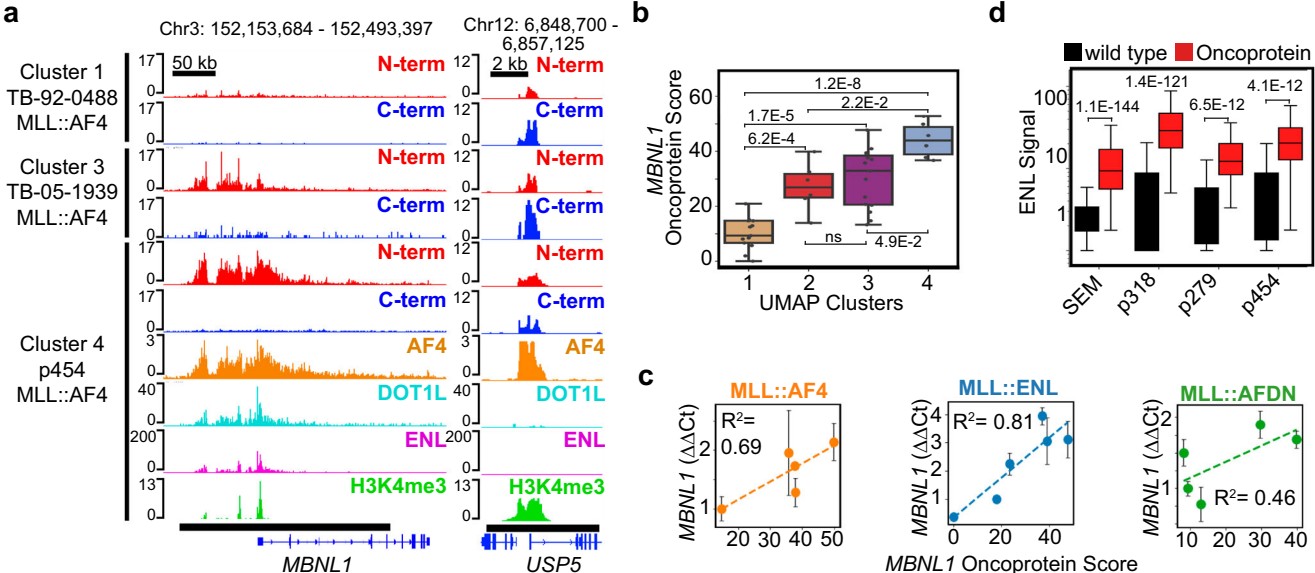

**Fig. 2 | MLL oncoprotein levels scale with target gene expression. a** Genome Browser tracks show the difference in MLL N-terminal (red) and C-terminal (blue) signals in three *MLL::AF4* bearing leukemias, over the most common oncoprotein-target locus: *MBNL1* (bottom black bar = oncoprotein-target peak). The *USP5* promoter is an example of a wild-type MLL target site (bottom black bar = wild-type MLL peak). AF4 (brown) DOT1L (cyan) ENL (magenta) and H3K4me3 (green) signal is also shown for the p454 sample. **b** Oncoprotein scores over *MBNL1* (quantified over the bottom-black bar in (**a**)) are representative of the average oncoprotein scores in each UMAP cluster; *p* values were computed using a two-tailed independent samples t-test; Cluster 1 *n* = 13 samples, Cluster 2 *n* = 6 samples, Cluster 3 *n* = 15 samples, Cluster 4 *n* = 5 samples; boxplot center lines = median, box limits = first and third quartiles, whiskers = 1.5 times the interquartile range (IQR). **c** The *MBNL1* oncoprotein scores are correlated with the relative expression of *MBNL1* as measured by qPCR; dotted lines indicate the regression and $R^2$ measures the fit to the data, error bars indicate the standard deviation of the mean of three qPCR biological replicates. **d** ENL is significantly enriched over the oncoprotein-target sites. *p* values were computed using a two-tailed independent samples t-test; *n* values are listed as wild-type site #, oncoprotein site #: SEM = 15611, 491; p318 = 22014, 246; p279 = 25672, 272; p454 = 18509, 438; boxplot center lines = median, box limits = first and third quartiles, whiskers = 1.5 times the interquartile range (IQR). Source data are provided as a Source Data file.

modified approach for calling MLL-oncoprotein-binding sites, we applied AutoCUT&Tag to profile DOT1L and ENL and AutoCUT&RUN to profile the AF4 fusion partner in the SEM *MLL*r cell line and three additional patient samples bearing *MLL::AF4* translocations. AF4 shows a very similar pattern of enrichment to the MLL N terminus over regions called as oncoprotein-target sites (Fig. 2a, see *MBNL1*), but is also enriched over many of the wild-type MLL binding sites (Fig. 2a, see *USP5*). AF4 does not effectively distinguish oncoprotein-target sites from wild-type sites in patient samples (Supplementary Fig. 3j). DOT1L is significantly more enriched at the oncoprotein-target sites than the wild-type sites in three out of four samples (Fig. 2a, Supplementary Fig. 3k), and ENL is significantly more enriched at oncoprotein-target sites in all of the samples we profiled (Fig. 2a, d). Wild-type MLL catalyzes Histone-H3-Lysine-4 trimethylation (H3K4me3) at gene promoters, and AutoCUT&Tag profiles of H3K4me3 do not effectively distinguish wild-type and oncoprotein-target sites, serving as a negative control (Fig. 2a, Supplementary Fig. 3l). We conclude, our modified approach for comparing the enrichment of the MLL N and C terminus is effective for calling oncoprotein-target sites. Co-occupancy with oncoprotein cofactors DOT1L and ENL can validate these calls, whereas co-occupancy with AF4 cannot.

## Lineage-specific and fusion-partner-dependent binding sites

By arranging samples according to the first two principal components, we find the MLL-fusion partners determine a large proportion of the variance in the oncoprotein-binding sites. Specifically, Principal Component 1 (PC1) primarily captures the variance in the binding sites of the MLL::AF4 and MLL::ENL fusion proteins and is tightly associated with the average oncoprotein scores in the ALL samples (Fig. 3a, Supplementary Fig. 4a). On the other hand, Principal Component 2 (PC2) captures the variance in the binding sites of the MLL::AF10 and MLL::AFDN fusion proteins in the AML samples

(Fig. 3a, Supplementary Fig. 4b). PC1 and PC2 are both associated with the magnitude of the oncoprotein scores in the MLL::AF9 samples, but to an intermediate degree (Fig. 3a, Supplementary Fig. 4a, b). When grouped according to the oncoprotein-fusion partners, the slope of the best-fit lines for PC1 versus PC2 matches the prevalence of each translocation in *MLL*r ALLs versus AMLs (Fig. 3a, b)[21]. This strongly suggests the MLL-fusion partner influences the genome-wide binding sites of the oncoprotein as well as the lineage specificities of the leukemias produced by the associated translocations.

Next, we identified oncoprotein-binding sites that contribute to the variance in PC1 and PC2 by rank ordering the sites according to the loading values (Fig. 3c, d). The oncoprotein target genes *GNAQ*, *TAPT1* and *FLT3* are among the top contributors to PC1 (Fig. 3c, Supplementary Data 3), whereas *SKAP2*, *SENP6*, *ZNF521* and *HOXA9* are among the top contributors to PC2 (Fig. 3d, Supplementary Data 3). The group of "highly sensitive" genes that a previous study found are rapidly downregulated in response to auxin-mediated MLL::AF9 degradation includes genes that contribute to both PC1 and PC2 (*MEF2C*, *MEIS1* and *SOCS2*) as well as PC1-specific genes (*CPEB2*) or PC2-specific genes (*SKIDA1*, *PBX3*, *HOXA9*; Supplementary Data 3)[22]. The oncoprotein-target genes in PC1 are enriched for Gene Ontology (GO) terms that include the regulation of hematopoiesis, myeloid cell differentiation, leukocyte differentiation, and immune responses (Supplementary Fig. 4c). The oncoprotein-target genes in PC2 primarily regulate developmental patterning (Supplementary Fig. 4d). To determine how the PC1-ALL-gene program and the PC2-AML-gene program are related to normal hematopoiesis, we plotted the average z-scores on a previously generated UMAP of single-cell RNA-sequencing data from lineage depleted bone marrow[23,24] (Fig. 3e–g). We find the PC1-ALL-gene program is strongly enriched for genes normally expressed in pro-B cells (Fig. 3f), whereas the PC2-AML-gene program is strongly

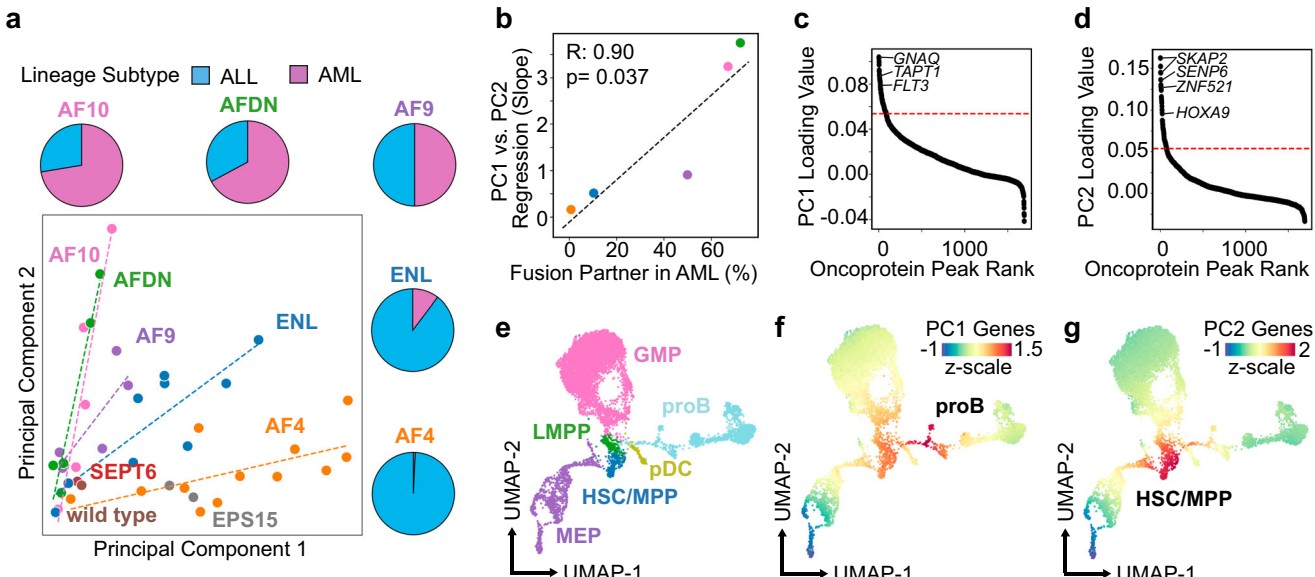

**Fig. 3 | At elevated levels the MLL oncoproteins are instructive of lineage-specific genomic binding sites. a** Principal Components 1 and 2 organize *MLL*r leukemia samples according to the oncoprotein-fusion partners. Dotted regression lines are included for samples bearing related translocations. Pie charts indicate the frequency of each MLL-fusion partner found in ALL versus AML[18]. **b** The slope of the MLL-oncoprotein regression lines from (**a**) are highly correlated with the frequency that each fusion partner is found in AML versus ALL. **c** The top 5% of sites (indicated by the red line) that contribute most to Principal Component 1 were identified by rank ordering the loading values. **d** same as (**c**) but for Principal Component 2. **e** UMAP from Otto et al. ref. 23 of single-cell RNA-seq data from 7439 cells from lineage-depleted bone marrow; HSC/MPP = Hematopoietic Stem Cell/Multipotent Progenitor, MEP = Megakaryocyte Erythroid Progenitor, pDC = pre-Dendritic Cell, proB = pro-B Cell, LMPP = Lymphoid-Myeloid-Primed Progenitor, GMP = Granulocyte Monocyte Progenitor. **f** UMAP embedding from (**e**) colored by the average z-score of genes from PC1. **g** UMAP embedding from (**e**) colored by the average z-score of genes from PC2.

enriched for hematopoietic stem cell and multipotent progenitor specific genes (Fig. 3g).

Our results suggest the MLL-fusion partner influences the lineage identity of the leukemia, which prompted us to examine this relationship more closely on specific oncoprotein target genes. The most frequent oncoprotein target genes *MBNL1*, *MEIS1*, *JMJD1C* and *MEF2C* all contribute to both PC1 and PC2, and the oncoprotein scores on these genes scale with the average oncoprotein score in each sample (Supplementary Fig. 4e–h, Supplementary Data 3). When the oncoproteins are highly abundant, PC1-specific oncoprotein-target genes, such as *FLT3* and *TAPT1*, are consistently bound by the B-ALL-enriched MLL::AF4 and MLL::ENL fusion proteins (Fig. 4a–c, Supplementary Fig. 4i). In comparison, only one MLL::AFDN sample and none of the MLL::AF10 samples had appreciable levels of the oncoprotein on the *FLT3* locus (Fig. 4b). Interestingly, the MLL::AF10 oncoprotein scores over the *TAPT1* locus scale with the overall oncoprotein levels in these samples, while the levels of MLL::AFDN on the *TAPT1* locus do not (Fig. 4c). This suggests *FLT3* is a lineage-specific oncoprotein-target gene whereas *TAPT1* is a fusion-partner-specific oncoprotein-target gene. By applying qPCR to a subset of these samples, we find the relative expression of *FLT3* is strongly correlated with the MLL::ENL oncoprotein scores over the *FLT3* locus but not the MLL::AF4 or MLL::AFDN oncoprotein scores (Fig. 4d). This indicates the *FLT3* locus is sensitive to the dosage of MLL::ENL but may become saturated and reach its maximum expression level when occupied by even low levels of MLL::AF4.

We also identified lineage-specific and fusion-partner-specific oncoprotein targets that contributed to PC2. The *ZNFS21* gene is critical for *MLL*r AMLs[25,26], and the MLL::AF10, MLL::AFDN and MLL::AF9 oncoprotein scores on the *ZNFS21* locus are positively correlated with the overall oncoprotein scores in the AML samples (Fig. 4e,f). In comparison, the *ZNFS21* locus was not bound by the oncoproteins in any of the MLL::AF4 B-ALL samples, and was only bound in two of the MLL::ENL samples (Fig. 4f). As expected, the relative expression of

*ZNF521* is positively correlated with MLL::AFDN, but not MLL::AF4 and only one of the MLL::ENL samples profiled by qPCR had detectable *ZNF521* expression (Fig. 4g). We conclude that in *MLL*r leukemias *ZNF521* is an AML-specific oncoprotein target gene.

The *HOXA9* locus also contributes to PC2, and the *HOXA9* oncoprotein scores scale with the overall oncoprotein levels in all the samples except those bearing *MLL::AF4* translocations (Fig. 4h, i, j, Supplementary Data 3). *HOXA9* was not called as an oncoprotein target in three out of six of the MLL::AF4 samples from UMAP Cluster 4, which had the highest overall oncoprotein levels (Supplementary Fig. 4j). Previous studies identified a subset of infant *MLL::AF4*-rearranged samples that have an exceptionally high risk of relapse and are characterized by low expression of genes in the *HOXA* cluster, and high expression of the homeobox transcription factor *IRX1*[8,27–29]. This gene expression pattern has also been reported in *MLL::AF4*-rearranged B-ALLs that went on to lineage switch[10]. We find that *IRX1* is a direct target of MLL::AF4 in the three samples from Cluster 4 in which *HOXA9* is not called as an oncoprotein-target gene (Supplementary Fig. 4k–m, Supplementary Data 2). When measured by qPCR, we find the expression of *HOXA9* and *IRX1* are generally mutually exclusive (Fig. 4j). The p279 sample is a B-ALL that went on to lineage switch in response to ALL-directed chemotherapy and is the only sample that expresses both *HOXA9* and *IRX1*. In p318, the AML that resulted from the lineage switch of p279, we found *HOXA9* is downregulated, whereas the expression of *IRX1* is maintained (Fig. 4j). Our data suggests there is a dichotomy in the direct regulation of *HOXA9* and *IRX1* by the MLL::AF4 oncoproteins which may contribute to lineage-switching events. Together, we conclude that at high levels of expression, the oncoproteins activate genes that influence the lymphoid versus myeloid identity of the leukemia (Fig. 4k).

### Oncoprotein dynamics and lineage switching
To characterize the epigenetic mechanisms underlying the lineage plasticity of *MLL*r leukemias, we performed a comprehensive

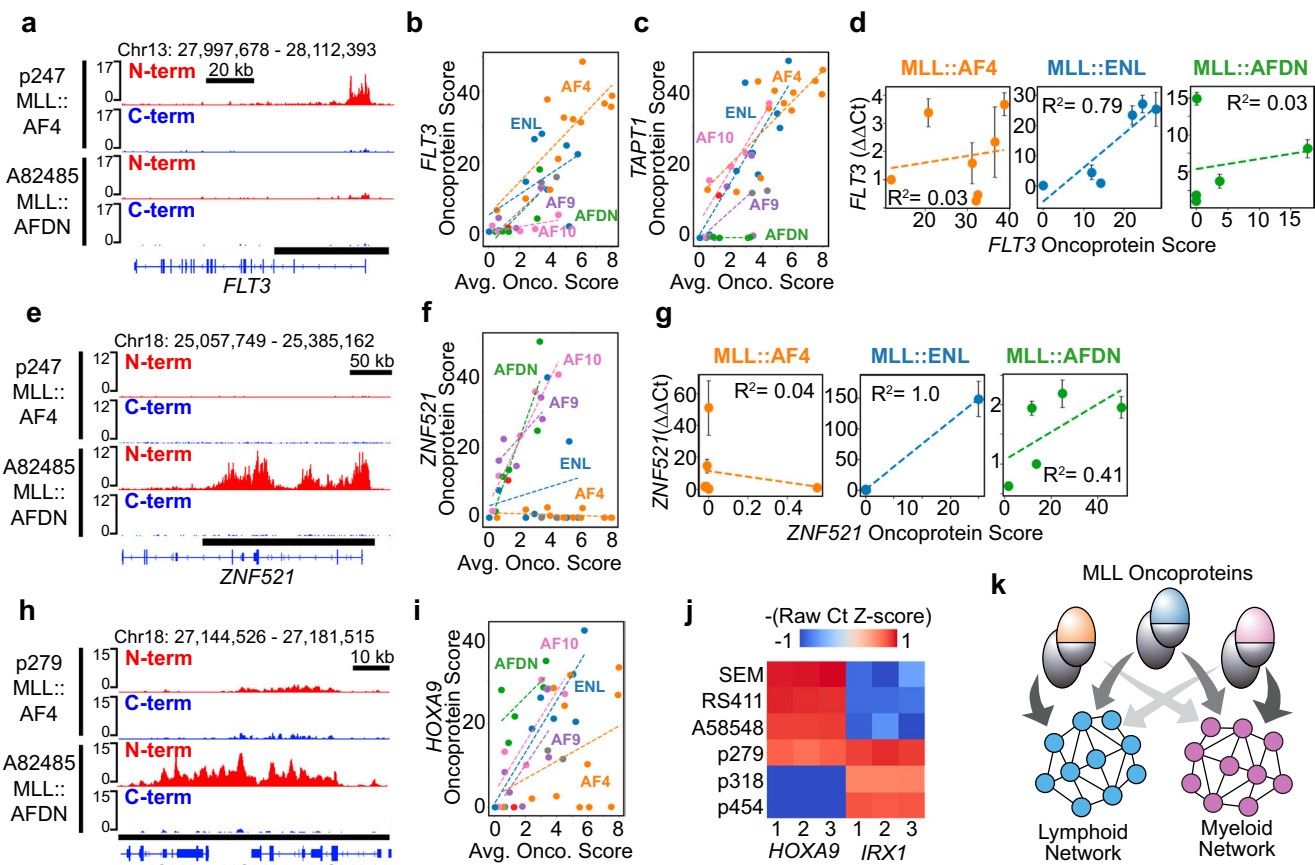

**Fig. 4 | MLL oncoproteins bind to lineage-specific and fusion-partner specific genomic sites. a** Genome Browser tracks show the MLL::AF4 oncoprotein binds the *FLT3* locus while the MLL::AFDN oncoprotein does not (bottom black bar = oncoprotein-target peak). **b** Scatterplot showing the *FLT3*-oncoprotein scores (quantified over the bottom black bar in (**a**)) relative to the average oncoprotein score in each sample. Samples are colored by the MLL-fusion partner and dotted-lines indicate the regression. **c** Same as (**b**) but showing *TAPT1* oncoprotein scores. **d** The MLL::ENL oncoprotein scores on *FLT3* are correlated with the relative expression of *FLT3* as measured by qPCR, while MLL::AF4 and MLL::AFDN levels are not correlated with expression of *FLT3*; dotted-lines indicate the regression and $R^2$ measures the fit to the data, error bars indicate the standard deviation of the mean of three qPCR biological replicates. **e** Same as (**a**) showing the *ZNF521* locus. **f** Same

as (**c**) but showing *ZNF521* oncoprotein scores. **g** The MLL::AFDN oncoprotein scores on *ZNF521* are correlated with the relative expression of *ZNF521* as measured by qPCR. Only one MLL::ENL sample expresses *ZNF521*. **h** Same as (**a**) showing the *HOXA9* locus, and a different *MLL::AF4*-rearranged sample in which *HOXA9* is not called as an oncoprotein target gene. **i** Same as (**b**) but showing *HOXA9* oncoprotein scores. **j** *HOXA9* and *IRX1* show mutually exclusive expression in *MLL::AF4* samples, except for the p279 sample that went on to lineage switch and give rise to the p318 AML sample. The Ct values from three qPCR biological replicates are shown. **k** The MLL-fusion partners direct the genomic occupancy of the oncoproteins to preferentially activate either lymphoid or myeloid lineage programs. Source data are provided as a Source Data file.

comparison of the oncoprotein-target genes in the patient-matched lineage-switching samples p279 and p318. We found that oncoprotein scores are generally reduced following relapse as AML (Fig. 5a), with a corresponding decrease in the mRNA expression of the oncogene (Fig. 5b). This finding is consistent with a previous report showing decreased oncogene expression during lineage switching in three out of four of the MLL::AF4-rearranged samples profiled by RNA-seq[10]. Next, we identified oncoprotein-target sites with the most dramatic differences between the p279 B-ALL sample and the p318 AML sample (Supplementary Fig. 5a). *BANK1*, *IGF2BP2*, *WDR66* and *DNTT* were among the B-ALL-specific oncoprotein-target genes (Fig. 5c, Supplementary Fig. 5a, Supplementary Data 4). As expected, these genes were expressed at significantly higher levels in the B-ALL sample prior to lineage-switching (Fig. 5d). Conversely, *FNDC3B*, *NR5A, and SIMC1* were among the genes identified as AML-specific targets (Fig. 5e, Supplementary Fig. 5a, Supplementary Data 4), and were dramatically upregulated in the AML sample (Fig. 5f). We conclude the expression levels and genome-wide localization of the MLL oncoproteins are dynamic and these changes are likely interdependent with the lineage identity of the leukemia.

We then compared the collection of B-ALL-specific and AML-specific oncoprotein-target genes to samples that were not involved in a lineage-switching event. The majority of the B-ALL-specific oncoprotein-target sites (64/66) were also identified in other samples (Fig. 5g, Supplementary Data 3). *BANK1*, *IGF2BP2*, *WDR66* and *DNTT* all contributed to the PC1-ALL-gene program (Supplementary Data 3), and expression of the B-ALL-specific oncoprotein-target genes is normally enriched in pro-B cells (Supplementary Fig. 5b). Interestingly, 51/82 of the AML-specific oncoprotein-target genes did not overlap with the oncoprotein-target genes in the other samples we profiled (Fig. 5g, Supplementary Data 4). This group of 51 oncoprotein-target genes, which we refer to as the "GMP-like program," is enriched for genes that are expressed in GMPs during normal hematopoiesis (Fig. 5h). By analyzing RNA-seq data previously collected from six *MLL::AF4*-rearranged samples that underwent a B-ALL-to-AML lineage-switching event[10], we found the B-ALL-specific oncoprotein-target genes are expressed at significantly higher levels in all six patient-matched B-ALL samples (Fig. 6a). Conversely, the GMP-like program is upregulated in all six of the AML samples, although this difference was not significant in one of the samples (Fig. 6b). We conclude that the

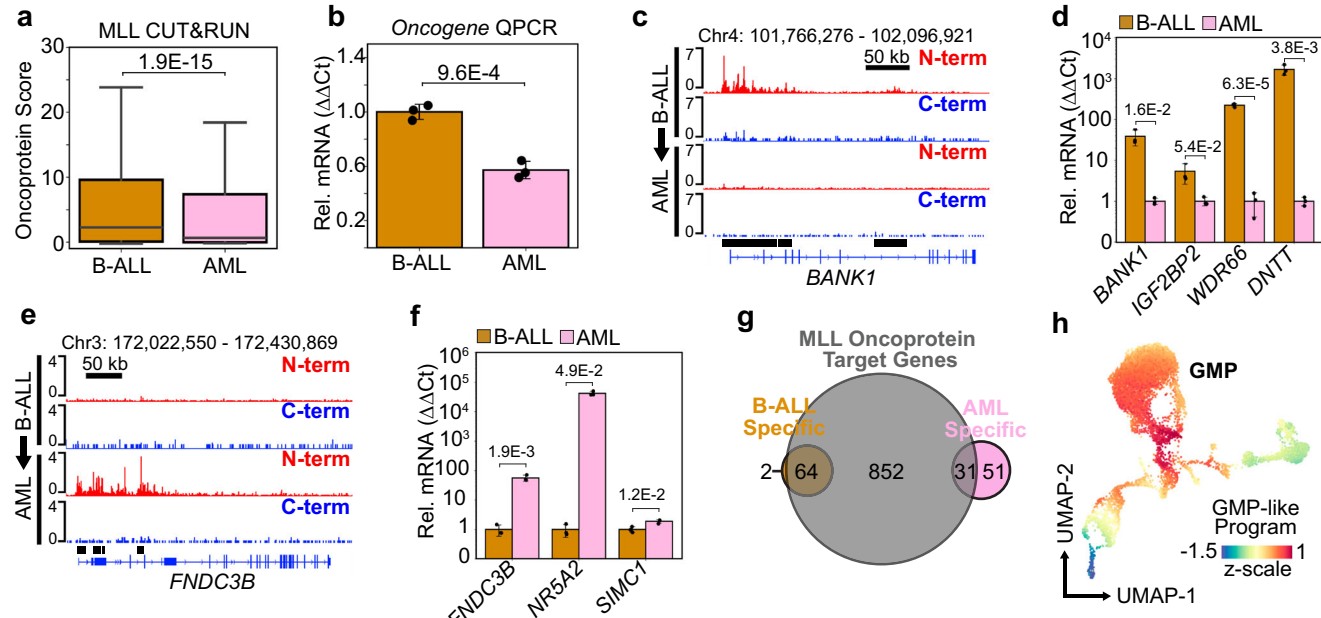

**Fig. 5 | The oncoprotein-binding sites change during B-ALL-to-AML lineage-switching events. a** Boxplot comparing the oncoprotein scores in a patient matched B-ALL and AML sample that lineage switched in response to B-ALL directed therapy. The $p$ value was computed using a two-tailed independent samples t-test; $n = 1692$ oncoprotein-target sites; boxplot center lines = median, box limits = first and third quartiles, whiskers = 1.5 times the interquartile range (IQR). **b** Oncogene expression is significantly reduced in the AML after lineage switching. Bar height is the average of three qPCR biological replicates; Error bars = standard deviation, and the $p$ value was computed using a two-tailed independent samples t-test; $n = 3$ qPCR biological replicates. **c** Genome Browser tracks showing the oncoprotein levels are elevated over the *BANK1* locus in the B-ALL sample prior to lineage switching. Bottom black bars indicate B-ALL specific oncoprotein target regions. **d** B-ALL specific oncoprotein target genes are expressed at significantly higher levels in the

B-ALL sample (gold) prior to lineage switching than the AML sample (pink). Bar height is the average of three qPCR biological replicates; Error bars = standard deviation; $p$ values computed using a two-tailed independent samples t-test; $n = 3$ qPCR biological replicates. **e** Genome Browser tracks showing the oncoprotein levels are elevated over the *FNDC3B* locus in the AML sample after lineage switching. Bottom black bars indicate AML specific oncoprotein target regions. **f** Same as (**d**) but for AML-specific oncoprotein target genes. **g** A Venn Diagram comparing the lineage-switching B-ALL-specific and AML-specific oncoprotein-target genes to all other oncoprotein-target genes. The AML group includes 51 non-overlapping genes referred to as the GMP-like program. **h** The UMAP embedding of healthy lineage-depleted bone marrow from Otto et al. ref. 23, colored by the average z-scores of the 51 non-overlapping AML-specific oncoprotein target genes from (**g**). Source data are provided as a Source Data file.

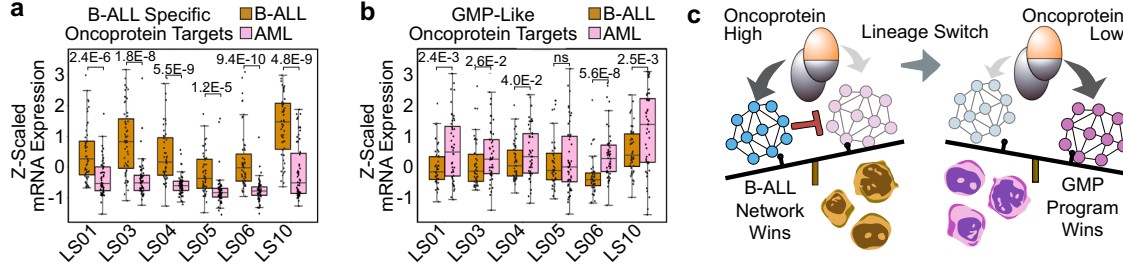

**Fig. 6 | MLL oncoprotein dynamics influence target gene expression during B-ALL-to-AML lineage-switching events. a** Boxplot comparing the relative expression of the B-ALL-specific oncoprotein target genes in a cohort of 6 B-ALL (gold) and AML (pink) patient-matched samples that underwent a lineage-switching event as described in Tirtakusuma et al. ref. 10. $p$ values were calculated using a one-sided Wilcoxon sign-rank test, $n = 66$ genes; boxplot center lines = median, box limits = first and third quartiles, whiskers = 1.5 times the interquartile range (IQR). **b** Same as

(**a**) but comparing the relative expression of the 51 genes in the GMP-like program, $n = 51$; boxplot center lines = median, box limits = first and third quartiles, whiskers = 1.5 times the interquartile range (IQR). **c** At high levels the MLL oncoprotein reinforces the B-ALL identify by activating a lymphoid program. During lineage switching the oncoprotein levels are reduced, and the oncoprotein activates a GMP-like program.

shift from direct activation of lymphoid-associated genes to activation of the GMP-like program is likely a recurring mechanism of oncoprotein-regulated lineage switching (Fig. 6c).

### Menin-inhibitor-resistant ENL-marked epigenetic lesions

A subset of *MLL*r leukemias initially respond to the menin inhibitor revumenib by downregulating oncoprotein-target genes, but for unknown reasons are resistant to treatment, and rapidly relapse[16]. One of the *MLL::ENL*-rearranged samples we profiled underwent a lineage-

switching event during treatment with the menin inhibitor. We compared the B-ALL and AML patient-matched samples (designated 148752 and 152985, respectively), and while 46 oncoprotein-binding sites were detectable in the B-ALL sample at diagnosis, treatment with the menin inhibitor reduced the genome-wide MLL N and C terminal signal to undetectable levels (Fig. 7a, Supplementary Fig. 6a, b, Supplementary Data 2). Similar to the p279/p318 samples, we observed a reduction in oncogene expression in the 152985 AML sample after lineage switching (Fig. 7b). However, while the oncoprotein became

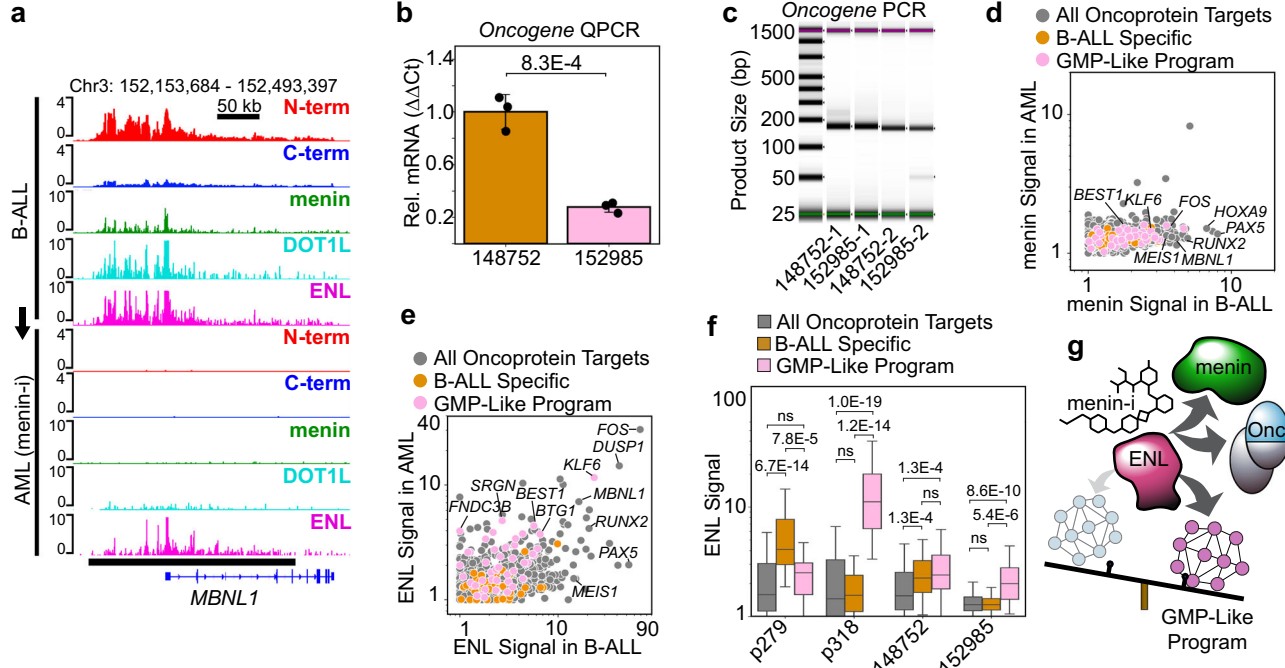

**Fig. 7 | ENL remains bound to MLL oncoprotein target genes during treatment with the menin inhibitors. a** Genome Browser tracks showing MLL, menin, DOT1L and ENL bound to the *MBNL1* locus in the B-ALL sample (top). In the patient-matched AML sample treated with the menin inhibitor only ENL remains bound (bottom). **b** Oncogene expression is significantly reduced in the AML after lineage switching. Bar height is the average of three qPCR biological replicates; Error bars = standard deviation, and the *p* value was computed using a two-tailed independent samples t-test; *n* = 3 qPCR biological replicates. **c** Reverse transcriptase PCR with two primer sets confirms the *MLL::ENL* oncogene is expressed in the AML treated with the menin inhibitor (expected size for 1 = 159 bp, and 2 = 150 bp). **d** Scatterplot comparing the normalized menin levels in the B-ALL and the AML sample treated with the menin inhibitor over an internal control group of oncoprotein-target sites (gray) the B-ALL-specific program (gold) and the GMP-like program (pink). **e** Same as (**d**) but showing the ENL signal persists over many of the oncoprotein-target genes in the AML sample. **f** Boxplot comparing ENL signal over an internal control group of oncoprotein-target genes (gray, *n* = 1595) the B-ALL-specific program (gold, *n* = 52) and the GMP-like program (pink, *n* = 45). *p* values were computed using a two-sided Mann-Whitney U Test with a Bonferroni correction for multiple hypothesis testing; boxplot center lines = median, box limits = first and third quartiles, whiskers = 1.5 times the interquartile range (IQR). **g** The menin inhibitor disrupts MLL oncoproteins, but ENL accumulation persists on oncoprotein targets in the GMP-like program. Source data are provided as a Source Data file.

undetectable in the menin-inhibitor-treated AML, the oncogene was still expressed, but at lower levels, as confirmed by reverse transcription PCR using two different primer sets (Fig. 7c). This suggests the menin inhibitor effectively eliminated the B-ALL cells by targeting the oncoprotein, yet the leukemia persisted in the AML state.

To explore how the AML cells expanded despite the targeted loss of the oncoprotein, we profiled the genome-wide localization of the oncoprotein cofactors menin, DOT1L and ENL by AutoCUT&Tag. As expected, menin, DOT1L and ENL were all bound to oncoprotein-target sites in the B-ALL sample (Fig. 7a, Supplementary Fig. 6a, b), and both menin and DOT1L were reduced to undetectable levels in the menin-inhibitor-treated AML (Fig. 7a, d, Supplementary Fig. 6a–c). Strikingly, ENL remained bound at appreciable levels over numerous canonical oncoprotein-target genes like *MBNL1* and *RUNX2* as well as genes in the GMP-like program such as *KLF6* (Fig. 7a, e, Supplementary Fig. 6a, b).

Next, we investigated the shifts in ENL binding during lineage switching by quantifying the relative levels of ENL over the B-ALL-specific genes and GMP-like program. We used the set of all oncoprotein-target intervals identified in the non-lineage-switching samples as an internal control. The B-ALL-specific and GMP-like gene signatures were originally identified by comparing the oncoprotein-target sites in the p279 and p318 samples. As expected, ENL is significantly more enriched over the B-ALL-specific genes in the p279 B-ALL sample and is significantly more enriched over the GMP-like program in the p318 AML sample (Fig. 7f). Interestingly, prior to treatment with the menin inhibitor, ENL was significantly enriched over both the B-ALL-specific genes and GMP-like program in the

148752 B-ALL sample, and during treatment, in the 152985 AML sample, ENL was only significantly enriched over the GMP-like program (Fig. 7f). The expression of B-ALL-specific genes *BANK1* and *WDR66* was reduced by ~50 and ~6 fold, respectively, in the 152985 AML sample (Supplementary Fig. 6d). The expression of *IGF2BP2* increased slightly in the 152985 AML sample (Supplementary Fig. 6d), and *DNTT* was unchanged, but was expressed at vanishingly low levels to begin with. In comparison, the expression of genes in the GMP-like program was either unchanged, such as *FNDC3B*, or reduced but to a lesser extent (Supplementary Fig. 6e). Specifically, *SIMC1* was reduced ~2 fold, *KLF6* was reduced ~6 fold and *NR5A2* was not expressed (Supplementary Fig. 6e, f). Together, we conclude the MLL oncoprotein induces accumulation of menin, DOT1L and ENL over the oncoprotein target sites. Once this epigenetic lesion forms, ENL can remain bound over a subset of oncoprotein-target sites, including genes involved in the GMP-like program, even after the oncoprotein is disrupted by the menin inhibitor (Fig. 7g).

Lastly, we examined whether the lineage-switching mechanism of menin-inhibitor resistance we discovered is related to other resistance mechanisms that have been described in the literature. Mutations in the *KMT2C* gene have been shown to lead to menin-inhibitor resistance in genetic models[17]. We found that in both AML samples that resulted from lineage-switching events the expression of *KMT2C* is down-regulated, however, this difference was only significant in the 152985 sample that lineage switched during treatment with the menin inhibitor (Fig. 8a). Next, we analyzed RNA-seq data from patient samples that were originally diagnosed as AML and developed resistance

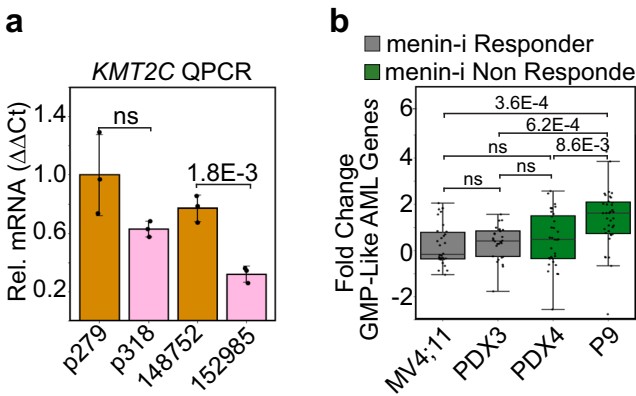

**Fig. 8 | *KMT2C* and GMP gene expression changes during menin inhibitor treatment. a** Expression of the *KMT2C* gene is significantly reduced in the lineage-switching sample treated with the menin inhibitor. Bar height is the average of three qPCR biological replicates; Error bars = standard deviation; p value computed using a two-tailed independent samples t-test; n = 3 qPCR biological replicates. **b** The menin-inhibitor-resistant AML patient P9 sample has significantly higher expression of the GMP-like program than two menin-inhibitor responder samples and one additional non responder profiled in Perner et al. 2023 (ref. 16). p values were computed using the two-sample t-test for independent samples. For comparisons of samples classified as non-responders to responders we used a one-sided test, otherwise we used a two-sided test. n is the number of GMP-like genes that had a significant fold change between the menin-inhibitor treated and untreated samples (padj. < 0.05): MV4;11 = 30, PDX3 = 28, PDX4 = 32, P9 = 39; box-plot center lines = median, box limits = first and third quartiles, whiskers = 1.5 times the interquartile range (IQR). Source data are provided as a Source Data file.

to the menin inhibitor in the absence of *MEN1* mutations[16]. As compared to the MV4;11 AML cell line and an AML patient-derived xenograft (PDX), which both demonstrated a prolonged response to menin inhibitors, the GMP-like program was significantly upregulated in the P9 AML patient sample that developed resistance (Fig. 8b). This indicates reductions in *KMT2C* expression and a shift toward the GMP-like program may be recurring mechanisms of epigenetic resistance to the menin inhibitors.

## Discussion

The lineage identities and therapeutic responses of *MLL*r leukemias are highly heterogeneous[2,5,7,8]. To characterize the molecular basis of this heterogeneity we applied Automated CUT&RUN to profile the oncoprotein-binding sites across 36 *MLL*r leukemia samples that span the age of onset and are representative of the diverse mutations and lineage subtypes that are observed in *MLL*r leukemias. We find that the oncoprotein levels and genome-wide binding sites are highly heterogeneous among patient samples and at high levels of expression the oncoprotein-fusion partner influences the oncoprotein-binding sites and is likely instructive of the lineage identity of the leukemia. By profiling patient samples that underwent B-ALL-to-AML lineage-switching events, we demonstrate that oncoprotein binding is dynamic. As compared to the patient-matched B-ALL samples, the oncogene expression and oncoprotein levels are generally reduced in the AML and the oncoprotein binding is shifted to a GMP-like program. The emergence of GMP-like cells during lineage-switching was previously suggested by single-cell transcriptomic and accessibility profiling[18], and our work demonstrates the transition to a GMP-like state is directly supported by the MLL oncoprotein. We also describe a B-ALL-to-AML lineage-switching event that occurred during treatment with a menin inhibitor and find ENL remains bound over the GMP-like oncoprotein-target genes. We find a similar GMP-like program is activated in a patient sample that presented as AML at diagnosis and rapidly developed resistance to the menin inhibitor in the absence of

*MEN1* mutations[16]. We propose that MLL-oncoprotein dynamics support the activation of a GMP-like program as a secondary relapse mechanism and that ENL likely contributes to the maintenance of this program in *MLL*r leukemias that are resistant to treatment with menin inhibitors.

Several mechanisms have been proposed to explain why the translocation partner genes *AF4* and *ENL* are much more common in B-ALLs whereas *AF10* and *AFDN* are more prevalent in AMLs[1,30,31]. Our results provide further evidence that these lineage biases are likely due to oncoprotein-fusion-partner dependent interactions with chromatin. We find the fusion-partner-dependent chromatin-binding events are more pronounced when the oncoprotein levels are elevated. For example, the MLL::AF10 oncoprotein consistently binds the *TAPT1* locus in AMLs while MLL::AFDN does not. In a subset of the infant B-ALLs we profiled, MLL::AF4 binds the *IRX1* locus at the expense of *HOXA9*, but MLL::ENL did not bind *IRX1* in any of the samples we profiled. This is consistent with a previous report that *IRX1* is expressed much more frequently in infant B-ALLs harboring the *MLL::AF4* rearrangement than the *MLL::ENL* rearrangement[8]. Several observations suggest the MLL-fusion partner acts interdependently with the lymphoid-versus-myeloid lineage context to direct the genome-wide localization of the oncoprotein. Our principal component analysis identified genes that are preferentially bound by MLL::AF4 and MLL::ENL in the lymphoid lineage (e.g., *FLT3*) and MLL::AF10 and MLL::AFDN in the myeloid context (e.g., *ZNF521*). Four of the samples we profiled did not conform to the fusion-partner-specific lineage biases. In the *MLL::AF10*-rearranged infant B-ALL and the *MLL::ENL*-rearranged AML we observed very low levels of the oncoprotein (Cluster 1 in Fig. 1d–f). Similarly, in the *MLL::AF4* and *MLL::ENL* lineage-switching samples, we observed the oncogene and oncoprotein were depleted in the AML sample after relapse. This suggests the AF4, ENL, AF10 and AFDN fusion oncoproteins require the appropriate lineage context to thrive, and efficiently initiate and reinforce the corresponding lineage subtype of leukemia.

The *MLL::AF9* mutation is found in roughly equivalent numbers of B-ALLs and AMLs[21]. We see MLL::AF9 accumulates over the *FLT3* and *TAPT1* B-ALL-associated target loci in AMLs (Fig. 4b, c), but the levels are below the threshold to be called as oncoprotein-target genes (Supplementary Data 2). In an AML induction model[22], MLL::AF9 binding is required to activate genes that contribute to both the PC1-B-ALL program as well as the PC2-AML program. Normally, AF9 is found in both the DOTCOM and SEC transcriptional complexes[32,33], and perhaps this broad interaction potential also underlies the lineage promiscuity of the *MLL::AF9* translocations.

The p279/p318 sample we profiled before and after lineage-switching provides a unique paradigm to consider the lineage-specific roles of the MLL oncoproteins. In addition to changes in the expression levels, the oncoprotein also binds to different sites in the B-ALL and AML samples. At diagnosis the oncoprotein is directly bound to genes in the B-ALL lineage program, and it is possible the lower levels of expression in the AML facilitate the non-canonical binding pattern over the GMP-like program. This provides a compelling case for a lineage-interdependency model in which the oncogene expression levels and oncoprotein-binding site selection both contribute to the lineage biases of specific *MLL* translocations.

Lineage switching could result from the trans-differentiation of B-ALL cells to an AML identity, or from the selective outgrowth of a latent multipotent cell type that retains both B-ALL and AML potential. A strong shift in the oncoprotein levels might support a trans-differentiation event. However, multiple lines of evidence suggest a selective outgrowth model is more likely. First, our comparison of oncoprotein-binding sites did not readily distinguish the B-ALL lineage-switching samples from the other B-ALL samples we profiled. Second, whole-exome sequencing of samples before and after lineage switching revealed a paucity of shared mutations[10]. And third, studies

using single-cell RNA-seq identified a rare population of myeloid-biased cells in lineage-switching samples[18]. The latent multipotent progenitor model predicts that while the MLL::AF4 oncoprotein was thriving in the p279 B-ALL cells, the ALL-directed chemotherapy was effectively able to eliminate them. Then, the p318 AML sample arose from the same latent population that gave rise to the p279 B-ALL cells.

Single-cell-genomic profiling of hematopoietic stem and progenitor cells indicates the lymphoid lineage arises from a restricted progenitor subtype called a Lymphoid-Myeloid-Primed Progenitor (LMPP) that also generates GMPs[23,34,35]. Perhaps this LMPP population acts as the cell of origin for lineage-switching *MLL*r leukemias, and provides an opportunity for the oncoprotein to induce or reinforce the GMP-like program as a secondary relapse mechanism. Future studies are necessary to elucidate any hematopoietic stem and progenitor cell-type-specific activities of the oncoproteins and determine the cellular context(s) where the oncoprotein activates the GMP-like program.

Intensive genetic and biochemical characterization has demonstrated menin, DOT1L and ENL act as transcriptional cofactors of the oncoprotein[32,33,36–41]. Numerous compounds targeting each of these interactions have shown promise in preclinical models[14,15,22,42–46], and phase I clinical trials indicate menin inhibitors are effective for treating a subset of relapse refractory patients with *MLL*r leukemia[5]. Several examples of genetic mechanisms that lead to lineage switching or acquired resistance to menin inhibitors have been described[10,16,17]. Our results suggest MLL oncoproteins also promote resistance to targeted therapies through epigenetic mechanisms. During lineage switching the oncoprotein activates a secondary GMP-like program, and ENL can persist on these genes even after the oncoprotein is effectively eliminated through treatment with a menin inhibitor. Although genetic alterations might have contributed to resistance in these samples, our results indicate that ENL likely maintains the activation of numerous MLL-oncoprotein-target genes and warrant investigation of menin inhibitor and ENL inhibitor combination therapies.

## Methods

### Patients
Patient samples were obtained from member COG institutions, Texas Children's Hospital and the St. Jude Children's Research Hospital in accordance with the Declaration of Helsinki. Written consent was obtained from all patients to permit the use of their de-identified samples in medical research. In the event the patient was a minor, written consent was obtained from the parent/guardian. For seven patient samples consent for sharing primary DNA sequencing data was not explicitly given, and to protect any patient-specific sequence information the data for these patient samples is provided as hg38 aligned files only. The studies were overseen by the institutional review boards at the Fred Hutch Cancer Center (IR protocol 9950), St. Jude Children's Research Hospital and Texas Children's Hospital. None of the patients received compensation for their inclusion in this study.

### Sample Information
The primary CD34 + HSPC control sample, as well as the four *MLL*r cell lines, 1 primary-patient ALL, 5 primary-patient AMLs, and 2 primary-patient MPALs were profiled in a previous study[19]. In this study, we profiled an additional 8 primary *MLL*r AML samples from a biobank maintained by the Meshinchi laboratory at the Fred Hutchinson Cancer Center and 7 infant *MLL*r ALL leukemias from the St. Jude's Children's Research Hospital. We also profiled 2 pediatric *MLL*r ALLs and 6 infant *MLL*r ALLs from the Texas Children's Hospital, including the samples that underwent a lineage-switching event in response to B-ALL directed treatment and during treatment with the menin inhibitor, as well as the two corresponding patient-matched AML samples collected upon relapse. Last, we profiled two infant *MLL*r B-ALL samples from Seattle Children's Hospital that were collected from the same donor prior to a

lineage-switching event. All sample transfers were performed in accordance with institutional regulatory practices. These samples were collected from the whole blood of patients with greater than 80% blast counts by performing a Ficoll separation to remove red blood cells and neutrophils and the leukemia enriched samples were then cryopreserved. The specific *MLL* chromosomal translocation in each sample was determined using either whole-genome sequencing, targeted capture sequencing, or using standard cytogenetic approaches as previously described[9,47]. Expression of the oncogene was confirmed by qPCR in 19 of the samples using primers targeting the minimal exon junction between *MLL* and the fusion-partner genes. Sample information is provided as Supplementary Data 1.

### Antibodies
For AutoCUT&RUN profiling of MLL oncoproteins, we used the strategy we previously described in Janssens et al. Each sample was profiled using two antibodies targeting the MLL N terminus, MLL N1 (mouse monoclonal anti MLL used at a 1:100 dilution; Millipore, clone N4.4, cat. no. 05-764; https://scicrunch.org/resolver/RRID:AB_309976) and MLL N2 (rabbit monoclonal anti-MLL used at a 1:100 dilution; Cell Signaling Technology, clone D2M7U, cat. no. 14689; https://scicrunch.org/resolver/RRID:AB_2688009) as well as two antibodies targeting the MLL C terminus, MLL C1 (mouse monoclonal anti MLL used at a 1:100 dilution; Santa Cruz Biotechnology, clone H-10, cat. no. sc-374392; https://scicrunch.org/resolver/RRID:AB_10988264), and MLL C2 (mouse monoclonal anti MLL used at a 1:100 dilution; Millipore, clone 9-12, cat. no. 05-765; https://scicrunch.org/resolver/RRID:AB_309977). To profile the occupancy of the oncoprotein fusion partner AF4, we used two antibodies: (1) mouse monoclonal anti AF4 used at a 1:50 dilution; MyBioSource, cat. no. MBS190886, (2) rabbit polyclonal anti AF4 used at a 1:50 dilution; Thermo Fischer Scientific, cat. no. PA5-77068; RRID:AB_2720795. Reads from CUT&RUN profiles using the two AF4 antibodies were then pooled for downstream analysis. For all mouse primary antibodies, we also included a subsequent incubation with the secondary rabbit anti-mouse IgG antibody (used at a 1:100 dilution; Abcam, cat. no. ab46540; https://scicrunch.org/resolver/RRID:AB_2614925); this secondary serves as an adapter to ensure efficient binding of pA-MNase. This rabbit anti-mouse IgG antibody was also run in the absence of a primary antibody as an IgG negative control for each sample.

For AutoCUT&Tag profiling of the MLL-oncoprotien transcriptional cofactors we used Rabbit anti menin at a 1:50 dilution (Bethyl, cat. no. A300-105A); rabbit anti DOT1L at a 1:50 dilution (Cell Signaling Technology, cat. no. 90878S); and rabbit anti ENL at a 1:50 dilution (Cell Signaling Technology, cat. no. 14893S). For AutoCUT&Tag profiling of H3K4me3 we used Rabbit anti H3K4me3 at a 1:50 dilution (Active Motif, cat. no. 39159). To increase the number of pA-Tn5 molecules tethered to each antibody target site, all CUT&Tag reactions included the secondary antibody Guinea Pig anti-Rabbit IgG (1:100, antibodies-online, cat. no. ABIN101961).

### DNA sequencing and data processing
AutoCUT&RUN and AutoCUT&Tag sample processing was performed by the Fred Hutch Cancer Center Genomics Shared Resources Facility according to previously published protocols[19,20], available on the protocols.io website (dx.doi.org/10.17504/protocols.io.ufeetje and dx.doi.org/10.17504/protocols.io.bgztjx6n). We used the Agilent 4200 Bioanalyzer to assess the AutoCUT&RUN libraries. Up to 96 samples were pooled at equimolar concentrations and sequenced on a NextSeq 2000 instrument with a P2-100 flow cell by the Fred Hutch Cancer Center Genomics Shared Resources Facility. This yielded 5–10 million 2 ×50 bp paired end reads per sample. To remove adapter sequences, we preprocessed the reads using cutadapt version 2.9 with parameters -j 8 --nextseq-trim 20 -m 20 -a AGATCGGAAGAGCACACGTCTGA ACTCCAGTCA -A AGATCGGAAGAGCGTCGTGTAGGGAAAGAGTGT -Z.

Paired-end reads were then aligned to the UCSC hg38 human genome build using Bowtie2 version 2.4.4 with parameters --very-sensitive-local --soft-clipped-unmapped-tlen --dovetail --no-mixed --no-discordant -q --phred33 -I 10 -X 1000.

To generate the MLL N-terminal and C-terminal bed files for each sample, we first removed duplicate reads from individual replicates and then pooled and sorted the remaining reads from the two MLL N-terminal samples into a single file, and the two MLL C-terminal samples into a second file. These combined MLL N-terminal and MLL C-terminal bed files were then used to generate coverage normalized bedgraph and bigwig files.

### Identification of MLL target sites

We began by assembling a list of all MLL target sites and called peaks on the MLL N and C terminal files. We called peaks with SEACR[48] version 1.3 using the "stringent", "non" normalized settings with a false-discovery rate (FDR) threshold of 0.01. We then assembled a master MLL peak list by concatenating and merging all the individual MLL N and C terminal peak sets. To remove non-specific background peaks and repetitive elements, we removed features that were also called as peaks in the IgG negative control samples. Specifically, we combined the IgG negative control reads from samples processed on the same 96 well plate, and called peaks using the "stringent", "non" normalized settings with a variable FDR threshold ranging from 0.001 to 0.0001 to avoid calling more than 2500 IgG peaks for any group of samples. We then removed any of the MLL peaks that overlapped with the peaks called on the IgG sample set.

### Comparison of MLL oncoprotein target sites between samples

To call MLL oncoprotein target sites, we first quantified the number of base pairs from the N-terminal and C-terminal signal of each sample that overlap with all the combined MLL peak-genomic intervals. We then used these values to assign an "oncoprotein score" to each interval. Specifically, a Bayesian model was implemented, assuming two Multinomial distributions for the C-terminal and N-terminal reads. Each distribution was modeled with a uniform prior, indicating no preconceived knowledge of the distribution of reads into the peaks. Under this model, the fraction of C-terminal reads assigned to each peak (denoted as $p_C$) and the fraction of N-terminal reads assigned to each peak (denoted as $p_N$) were derived from the posterior distribution, assuming a Dirichlet distribution based on the count of reads (either N-terminal or C-terminal) in each interval plus a vector of ones. This implies the ratio between the two, $r_{NC}$, can be obtained as $\log_2(p_N/p_C)$; this value is referred to as the N/C score.

To estimate the mean and standard deviation of the N/C score for each interval, Monte Carlo sampling was performed. A total of 1000 samples (N = 1000) were drawn from this posterior distribution and the mean ($\mu_i$) and standard deviation ($s_i$) were calculated for each interval. These statistics provided the necessary information to compute a z-score and p-value for the N/C score of each genomic interval. The resulting p-values were then corrected for multiple testing using the Benjamini/Hochberg correction method, using the fdrcorrection function from the statsmodels Python package. This procedure offers a rigorous approach to statistically measure the differential distribution of N-terminal and C-terminal CUT&RUN reads across the genomic intervals of interest. For each sample, intervals with an N/C score greater than 1.75 and a p-value less than 0.00001 were called as oncoprotein target sites and these sites were combined to obtain a master list of 1692 oncoprotein target sites across all samples. We then calculated the "oncoprotein score" for all 1692 intervals in each sample by multiplying the N/C score by the $-\log_{10}$ of the p-value plus a pseudo count of 1E-10. The average oncoprotein score was then calculated by taking the average of these values for all 1692 sites.

### Quantitative PCR

Total RNA extraction was performed using the Qiagen RNeasy Plus Mini Kit using 1 million whole cells as input. We eluted the sample in 15 uL of nuclease free water, and found the total amount of RNA obtained did not exceed 1 µg. Reverse transcriptase (RT) reactions included 8.25 µL of the total RNA, 0.5 µL of Oligo(dT) for a total of 25 picomoles, 0.5 µL of Random hexamers for a total of 25 picomoles, 1 µL of dNTP Mix (10 mM each), 3 µL of 5X RT Buffer, 1 µL of Thermo RiboLock RNase Inhibitor, 0.75 µL of Maxima Reverse Transcriptase. The reaction was carried out on a thermocycler with a heated lid as follows: 10 min @ 25 °C, 30 min @ 50 °C, 5 min @ 85 °C hold @ 12 °C. After the RT reaction, samples were diluted 1:3 in nuclease free water. Samples were then diluted to obtain equal concentrations of total cDNA based on an initial assessment of the relative enrichment of the normalizer genes *ACTB* and *GAPDH*. To minimize the errors introduced by primer specific amplification efficiencies, total cDNA was normalized such that equal template concentrations were used for all subsequent reactions.

Primers for qPCR were designed using the IDT Primer Quest software and modified manually by adding nucleotides in the 5' direction to ensure a melting temperature of 58 °C. The sequences of all the primers used for qPCR are provided as Supplementary Data 5. All qPCR reactions were performed using the Applied Biosystems SYBR Green PCR Master Mix (Thermo Fischer Scientific cat. no. 4309155). All assays were performed as biological triplicates in 10 µL reactions containing 5 µL of PCR Master Mix, 3.585 µL of ddH2O, 0.415 µL of Forward and Reverse primers premixed to a concentration of 10 µM each. 1 uL of total cDNA was added to each reaction. qPCR reactions were carried out on the QuantStudio5 qPCR instrument named "Paul" available through the Fred Hutch Cancer Center Shared Resources. QPCR reactions were run for 40 cycles with the standard thermal profile with the exception that the annealing temperature and minimal temperature for the melt analysis were adjusted to 58 °C. Source data including all raw Ct values are provided in the Source Data file.

### Comparisons to RNA-seq

To examine the relative RNA expression in a cohort of six lineage-switching samples, the previously determined RPKM values for all the available B-ALL-specific and GMP-like oncoprotein-target genes in the LS01, LS03, LS04, LS05, LS06 and LS10 primary and relapsed patient samples were pulled from Supplementary Table 2 of Tirtakusuma et al.[10]. We then transformed these RPKM values according to the z-scale of each gene across all 12 samples, and compared the differences in z-scaled expression for B-ALL-specific and GMP-like oncoprotein-target genes between primary (B-ALL) and relapsed (AML) patient samples.

To determine how the expression of the GMP-like *MLL*r leukemia program genes changed in response to revumenib, we pulled all the corresponding log2FoldChange values that were available in the "suppTab.9_RNAseq_DEGs_non-genet" of Perner et al.[16] and were below an adjusted p value cutoff of 0.05. We compared these log2Fold-Change values from the MV4;11 sample treated with 1 µM SNDX5613 (Revumenib) to the PDX3, PDX4 and Patient 9 samples.

### Statistics

Comparisons of the average oncoprotein scores and the gene-specific oncoprotein scores between the four clusters of *MLL*r leukemia samples were performed using a two-tailed independent samples t-tests. To account for multiple comparisons and control the family-wise error rate, a Bonferroni correction was applied. This correction involved dividing the conventional alpha level (0.05) by the number of comparisons being made. Only p-values that met the corrected threshold for significance were considered statistically significant.

Comparisons of the relative expression of the MLL oncogenes as well as the MLL oncoprotein target genes, as measured by qPCR, were

performed using a two-tailed independent samples t-test. These values were not adjusted for multiple hypothesis testing.

Comparisons of the AF4, DOT1L, ENL and H3K4me3 signals between the MLL wild-type and MLL oncoprotein binding sites were performed using a two-tailed independent samples t-test and these values were not adjusted for multiple hypothesis testing.

To compare the RNA-seq gene expression values, we assumed a non-normal distribution and employed the Mann-Whitney-Wilcoxon test, a non-parametric equivalent to the independent samples t-test. This was performed as a one-sided test because in all cases our hypothesis was that the RNA expression levels would change in a manner that was concordant with the changes we observed in the MLL oncoprotein occupancy during lineage switching or the development of resistance to menin inhibitors. Given multiple pairwise comparisons, we adjusted for the increased risk of Type I error by applying the Bonferroni correction. Specifically, the significance level for each test was set by dividing the conventional alpha level (0.05) by the number of pairwise comparisons. All statistical analyzes were conducted using the add_stat_annotation function in Python, which automates the application of the specified statistical test and adjustment across the defined box pairs.

### Reporting summary

Further information on research design is available in the Nature Portfolio Reporting Summary linked to this article.

## Data availability

The CUT&RUN and CUT&Tag sequencing data that was generated for this study has been deposited into the Gene Expression Omnibus and is available under the accession code GSE252378. For seven of the patient samples we profiled (p160, p179, p186, p214, p247, p279 and p318) the raw sequencing data are not publicly available and will not be shared because patients did not consent to genomic data sharing. For these samples the aligned bed and bigwig files are available as part of the GSE252378 dataset, and these bed files are also available through Zenodo[49] https://doi.org/10.5281/zenodo.13791761. The raw sequencing data for all the other samples described in this study is publicly available. The additional CUT&RUN data analyzed in this study was generated by Janssens et al.[19] and the raw paired-end fastq files are available through the Gene Expression Omnibus under accession code GSE159608. The remaining data are available within the Article, Supplementary Information or Source Data file. Source data are provided with this paper.

## Code availability

The code used for processing CUT&RUN and CUT&Tag data (e.g., removing duplicates and combining replicates) as well as the pre-processed data tables, and the python jupyter notebooks used for figure generation are available at: https://github.com/DerekJanssens/MLL_oncoprotein_levels_NatComms and are also published on Zenodo[49] https://doi.org/10.5281/zenodo.13791761.

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

## Acknowledgements

We thank Christine Codomo, Terri Bryson and Jorja Henikoff for technical assistance, Matthew Fitzgibbon for bioinformatics support and Phil Corrin and Merari Santa-Carbajal and the rest of the Fred Hutch Genomics Shared Resources for performing AutoCUT&RUN and Auto-CUT&Tag as well as sequencing and data processing. We thank J. Stevens, as well as the Seattle Children's Hospital's Department of Anatomic Pathology and TTS Leukemia/Lymphoma Steering Committee, for assistance in tissue collection and research coordination. This work was supported by the Howard Hughes Medical Institute (S.H.), a Postdoctoral Fellowship from the Hartwell Foundation (D.H.J.) and a grant from the National Institutes of Health: NHGRI R01 HG010492 (S.H.). C.G.M. is supported by NCI R35 CA297695 and P30 CA021765 and the American Lebanese Syrian Associated Charities of St. Jude Children's Research Hospital.

## Author contributions

D.H.J. and J.F.S. conceived the study. D.H.J., W.W., and Y.X. performed the experiments; D.H.J., M.D., and D.J.O. performed data analysis. D.K., C.G.M., J.S.Y., and S.M. provided critical materials; D.H.J. and M.D. wrote the manuscript; D.H.J., S.H., J.F.S., K.A., and C.G.M. reviewed and edited the manuscript, and all authors approved the manuscript.

## Competing interests

The authors declare no competing interests.
