## [Transparent Peer Review file · Nature Communications]

MLL oncoprotein levels influence leukemia lineage identities

Corresponding Author: Dr Steven Henikoff

Version 0:

Reviewer comments:

Reviewer #1

(Remarks to the Author)

In this manuscript, Janssens et al. profiled the KMT2A-r (i.e., MLL-r) oncoprotein target genes in 34 samples (4 cell lines, 12 infant patients, and 18 Ped/Adult patients) leukemic blast using a state-of-the-art autoCUT&RUN method. The ratios between probing the KMT2A-N' vs C' signals allow enrichment of gene loci potentially bound by the KMT2A-r (only has the N'). The authors performed various clustering and principal component analyses and summarized an "oncoprotein score" that correlates with fusion partners (AFF1, ENL, AF9, etc.) and leukemia types (AML vs. B-ALL). They also utilized a B-ALL-to-AML lineage-switching sample to derive a panel of "non-canonical AML program" (42 genes). Finally, they observed the enrichment of "non-canonical AML program" in 2 MENIN-inhibitor resistant patients. Overall, the authors suggest the KMT2A-r oncoprotein binding targets can be dynamically (and epigenetically) shifted during leukemia progression and lineage-shift.

While the concept of this manuscript is highly interesting, there are several concerns requiring further validation. The main issues are the overall correlative analysis and the n = 1 lineage-switching sample, which make the "conclusion" more like an "assumption". Here are my specific comments:

1. "These differences in oncoprotein scores suggest that among leukemias bearing the most common KMT2A translocations the levels of the oncoprotein are extremely heterogeneous." – Ideally, an RNA-seq or a Western blot of the KMT2A-r oncoprotein expression in these samples should be presented to cross-compare with the "oncoprotein score" defined in this study.
2. The authors selected GNAQ to present in fig. 1e. However, how GNAQ was selected was not presented in the manuscript. A statistical analysis of 34 samples divided into Clusters 1, 2, 3, and 4 for their oncoprotein binding signal at GNAQ is necessary for evaluation. Similarly in Fig. 2, the selection of FLT3 and ZNF521 should include more statistical rationale derived from the dataset.
3. Fig. 2f and 2g: The PC1 Gene Ontology (GO) (Fig. 2f) related to blood lineages is expected. However, the PC2 GO (Fig. 2g) related to other tissue development (skeletal, etc.) was not readily connected to these leukemia samples. Can the authors speculate as to what could be the relevance of these GOs to the examined KMT2A-r leukemias?
4. "The group of "highly sensitive" genes that a previous study found are rapidly downregulated in response to auxin-mediated KMT2A-AF9 degradation include genes that contribute to PC1 (MEF2C, CPEB2 and MEIS1) as well as PC2 (SKIDA1, PBX3, HOXA9 and SOCS2; Supplementary Information)" – The information listed in the "Supplementary Information" will benefit from a clearer description or data presentation. For example, the MEIS1 and SOCS2 genes each were found in both PC1 and PC2 gene lists in the "Supplementary Information".
5. The current study derived a series of analyses based on 1 patient sample pairs that exhibited B-ALL-to-AML lineage-switching. The main concern is the reproducibility, as the conclusion was driven by n = 1. At least, the authors should speculate on additional factors that could bias the current conclusion, e.g., the "passive clonal selection" instead of "active switch" in this patient could also explain the lineage-switching. This also raises concerns as to whether the same switch-related signature could be observed in more patients experiencing lineage-switching during relapse.
6. The enrichment of "non-canonical AML program" (42 genes) in 2 MENIN-inhibitor resistant patients is intriguing. As a comparison, an examination of the "non-canonical AML program" (42 genes) in MENIN-inhibitor sensitive patients should be presented in the manuscript to evaluate the specificity of this signature to MENIN-i resistance.
7. The authors concluded "We propose that a dynamic shift in the binding sites of the oncoprotein activates an alternate leukemia program and this epigenetic mechanism promotes resistance to targeted therapies." – Indeed, there is no

“epigenetic” measurement (e.g., DNA/histone modification, ATAC-seq, chromatin looping, etc.) presented in this manuscript. Furthermore, the potential contribution of “genetic” alterations (i.e., mutations) was not examined in this study (so, this possibility could not be excluded). I recommend the authors refine the data interpretation to consider alternative possibilities.

Reviewer #2

(Remarks to the Author)

Janssens et al describe the analysis of a cohort of patient samples with KMT2A rearrangements that were analyzed with auto-Cut-and-Run for KMT2A-fusion binding. This is an extension of a previously published cohort interrogated with the same technique.

The key strength of this manuscript lies in the fairly large cohort and the use of primary patient samples. Key findings are the somewhat surprising inter-patient variability of binding sites, and the insights of how the fusion partner influences binding sites.

This work has substantial limitations:

- Detection of the fusion is very indirect, which reduces the confidence of the identification. An IgG control is used to informatically remove over 2500 peaks, then C-terminal peaks are subtracted from N-terminal peaks to indirectly compute fusion binding. Sweeping statement made based on 40 residual peaks in a single sample.
- It is not possible to look at the quality of the primary data for the Cut&Run as the GSE given seems to have a typo – the data set that comes up relates to astrocyte inflammation.
- The GSE extension for the previously published samples is functional. Analysis of the primary data showed it to be very noisy, with not a good replicate concordance. The published detailed analysis in the prior publication suggests that it is possible to detect meaningful patterns on a cohort basis. However, the high degree of technical variability raises concern that that some of the variability that is the primary finding of this study could be technical rather than biological. Of particular concern are conclusions that rely on just one or two samples.
- Given these concerns, more examples tracks of well described fusion targets should be shown, including the canonical well validated targets HOXA, Meis1, MEF2C. Representative examples should be shown in the main figure, with the cohort shown in the supplement to allow the reader to judge the quality and reproducibility of the Cut and Run without downloading the data.
- Several groups have reported a lower expression of HOXA cluster genes in KMT2A-AFF1 rearranged leukemias compared to other samples (Trentin et al. 2009, Stam et al. 2010 and Kang et al. 2012). Instead, these leukemias seem to frequently express IRX1. Do the authors find consistent differences in the HOXA cluster based on the fusion partner? How about IRX1?
- How do the results relate to ChIP studies on different fusions done in retroviral models?
(MLLL-AF4 and using subtraction ChIP (N - C)
How do they relate to ChIP studies in patient samples? It would be nice to see Venn diagrams that show overlap of the targets identified here compared to prior studies and other models.
- The prior publication showed that Cut&Run peak profiles at fusion targets recapitulated the broader peaks first reported by Tom Milne’s group. However, that analysis was done mostly on cell lines, and a small number patient samples. Does the broad versus narrow peak feature for N-only vs N+C peaks hold up in the larger patient sample cohort? Are there any differences based on fusion partner?
- There is a missed opportunity to conduct transcriptomic analysis in the same samples and integrate the two datasets. The authors use a published RNA-Seq experiment to interrogate transcriptional changes of specific loci – this cannot substitute for a careful and comprehensive characterization of fusion binding and matched gene expression.
- A correlation between fusion binding patterns and fusion partner was also shown in the prior publication.
- A single lineage switch sample is analyzed. It is very difficult to draw any firm conclusions from a single sample, particularly with a technique where the number of no-specific peaks exceeds the number of specific peaks by an order of magnitude, with raw data that is as noisy as what it uploaded in GSE 159608, and with the raw data for this sample unavailable. In order to draw any biological conclusions, this cohort should be expanded to include at least 4-5 additional samples. Finally, while auto-Cut&Run is suitable as a hypothesis generating technique for large numbers of samples, the more specific and precise ChIP is the more appropriate technique for low numbers of samples where greater weight falls on individual peaks being specific. Validation experiments in this extended (but still very small) lineage switch cohort should be performed by ChIP and integrated transcriptomic analysis of the same samples. While not likely available at a single institution, lineage switch is now common in an era of B-cell directed therapies, and it should be possible for the authors to put together a cohort of 5-6 samples total.
- The authors then relate the lineage switch pattern to a revumenib resistant RNA-Seq analysis from another publication. How do the authors know that these samples weren’t also lineage switch (or on the way there), and that, rather than the revumenib resistance, drove the gene expression profile? This part should be removed as the claims made by the authors are not supported by the experiment shown. In order to make such a claim, a much larger cohort, preferably pre and post revumenib, and linked Cut&Run and RNA-Seq profiling would need to be performed, ideally supported by functional validation what about this “non-canonical” AML program induces resistance. This endeavor seems outside the scope of a revision.
- There is inconsistent use of nomenclature, using official gene symbols for some fusion partners (KMT2A, AFF1), but not others (AF9, AF10, ENL).

Reviewer #3

(Remarks to the Author)

In this manuscript, the authors utilize Auto.CUT&RUN to examine KMT2A and KMT2A-oncoprotein localization in 34 KMT2Ar leukemia samples using Nterminal and Cterminal antibodies to differentially assess WT vs oncogene localization profiles. Using these data, the authors demonstrate heterogeneity in oncoprotein localization and observe switching of oncoprotein localization in the B-ALL to AML lineage switch sample included. With these data, the authors propose a model of oncoprotein shifting in KMT2Ar leukemias. I think the paper is very well written and clear, with exciting results that may be impactful to understanding molecular mechanisms in KMT2Ar-rearranged leukemias. I have the following points for clarification:

1) The authors detail using two N-terminal antibodies (recognize both the WT and oncoprotein) and two C-terminal antibodies in each of the 34 samples. From the methods description, I interpret the description to mean that the authors combine the AutoCUT&RUN data from each of antibodies, but I do not see any comparison of their specificity or signal. Do the individual N-terminal antibodies demonstrate similar patterns before combining the data and do the two C-terminal recognizing antibodies have similar patterns?

2) Related to the antibody choice and data presented, to calculate oncoprotein localization vs WT, the authors take N to C ratio. In Figure 1C, the authors demonstrate clear enrichment in the leukemia cells relative to the CD34+ WT cells. However, the browser tracks throughout the paper show clear N-terminal enrichment, but the C-terminal enrichment is very modest. Therefore, I wonder how robust the ratio calculation is; meaning that if the signal is very low, taking a ratio can be very skewed. While the example browser tracks throughout the manuscript have clear N-terminal enrichment and little C-terminal enrichment, is this true for all the ~1500 locations? Were there any examples where Cterminal was equivalent (or more) than N-terminal?

3) I realize that there are a lot of AutoCUT&RUN data presented in this manuscript, but the second half of the paper and some major conclusions rely on the single lineage switch sample that was profiled. While I appreciate that patient samples are difficult to obtain, and lineage switch samples are not common, having at least one other sample would really strengthen the conclusions drawn. For instance, in Figure 4A, are the non-overlapping AML peaks due to heterogeneity within lineage switch samples?

4) It was never clear to me if levels of KMT2A-oncoprotein levels were ever examined in the study. The model (Fig 3G) shows that low KMT2A-oncoprotein tips to AML; so in the lineage switch sample is KMT2A-oncoprotein lower relative to B-ALL samples? And are protein levels of the oncoprotein lower in AML relative to B-ALL in general?

Version 1:

Reviewer comments:

Reviewer #2

(Remarks to the Author)

Janssens et al describe the analysis of a cohort of patient samples with KMT2A rearrangements that were analyzed with auto-Cut-and-Run for KMT2A-fusion binding. This is an extension of a previously published cohort interrogated with the same technique. The key strength of this manuscript lies in the fairly large cohort and the use of primary patient samples. Key findings are the somewhat surprising inter-patient variability of binding sites, and the insights of how the fusion partner influences binding sites.

The revised manuscript is substantially improved with respect to identification of fusion target loci. The AFF1 angle (as NOT reflecting fusion status) is very interesting and helpful. Confirmation of the HOXA/IRX1 expression patterns and link to fusion partner is nice to see. The newer data is now accessible on GEO, and the quality is improved compared to the earlier data – likely reflecting some fine tuning and increased experience of the team. This is also nice to see.

Unfortunately, the initially submitted work has substantial limitations, which are only partially addressed in the revision.

1. A key concern that remains are results rely on just one or two samples. This concern has not been addressed. In the initial manuscript, generalized statements were made based on a single switch sample. Based on reviewer feedback, the team included two additional samples, and indeed found the response to be quite heterogenous. These findings underscore the critical importance of including more than once sample into key experiments. Unfortunately the authors then go on to examine ENL patterns in a single Menin inhibitor resistant sample, and again make sweeping conclusions based on a this

single sample. Ideally this would be removed from the work until a larger cohort can be put together. If the authors wish to retain these results, the conclusions have to be toned down substantially.

2. It should be mentioned that concern about conclusions based on a single sample were also voiced by reviewer 1.

3. The title “KMT2A oncoproteins induce epigenetic resistance to targeted therapies” makes no sense. If the results in the single Menin inhibitor resistant sample were to be taken at face value, then the fusion is actually gone from DNA. So something else maintains expression of genes required for transformation. The title should not refer to drug resistance since this is the not well substantiated part of the publication.

4. The presence of the elongation mark ENL on a subset of genes that retain expression speaks to the age-old conundrum of epigenetics – distinguishing the chicken from the egg. Does ENL reflect that these genes are being transcribed? Or is it driving transcription? The authors present no data to suggest it's the latter. In the absence of such data, it is much more likely that ENL reflects the GMP like state (well established to promote lineage switch) rather than drives it.

In light of these issues, the following statements need to be very much toned down:

- a. Title: KMT2A oncoproteins induce epigenetic resistance to targeted therapies (relies on a simple sample, also makes no sense)
- b. Abstract: “...can induce epigenetic lesions, marked by ENL, that support resistance to targeted therapies”. Interesting theory but not substantiated by the data.
- c. Discussion, first paragraph: the discussion of the GMP program during lineage switch needs to cite the work of Chen and colleagues that shows emergence of a GMP like cluster over the course of a lineage switch (Blood 2022).
- d. Discussion, first paragraph: “We propose that KMT2A-oncoprotein dynamics and the induction of “epigenetic lesions,” marked by ENL, play a critical role in the mechanisms that allow KMT2Ar leukemias to evade targeted therapies.” This needs to be toned down given that it is relying on a single sample and no functional experiments support ENL as the “driver” of resistance.
- e. Discussion, last paragraph: “our results point to ENL as linchpin of epigenetic resistance to menin inhibitors”. This conclusion is certainly not supported by correlative data in a single sample.

5. This reviewer asked for a supplemental figure that shows key target loci (HoxA, Meis, Mef2c) for the C- and N-terminal MLL (both replicates), which would allow readers to easily get a sense of the quality of the data that underlies the conclusions without having to download and reanalyse the GEO file. This was not done.

6. There is continued inconsistent use of nomenclature, using official gene symbols for some fusion partners (KMT2A, AFF1), but not others (AF9, AF10, ENL). The authors should pick one and stick with it (i.e. KMT2A-AFF1 / KMT2A-MLLT3 / KMT2A-MLLT10, or MLL-AF4 / ML-AF9 / MLL-AF10) etc.

Reviewer #3

(Remarks to the Author)

The authors have thoroughly addressed all the reviews from myself and the other reviewers. I think the addition of two more lineage switching lines has especially helped enhance the manuscript.

REVIEWER COMMENTS

We would like to thank all the reviewers for their thoughtful critiques and constructive feedback. Addressing all these concerns required additional experimentation and substantial revisions, and we hope the reviewers agree the result is a manuscript that now provides critical new insight into the epigenetic mechanisms of resistance in *KMT2Ar* leukemias. A major highlight of the revision is that we were able to obtain patient-matched B-ALL and AML samples that lineage switched during treatment with the menin inhibitor. As far as we know this is the first report of a lineage-switching event that occurred during treatment with the menin inhibitor. We applied our AutoCUT&RUN strategy to map the KMT2A-oncoprotein-binding sites as well as the cofactors menin, DOT1L and ENL in these samples and the results astonished us. Specifically, we find that in the AML sample the menin inhibitor effectively reduced the oncoprotein, menin and DOT1L to undetectable levels, however, ENL persisted on numerous oncoprotein-target genes and is enriched on the AML-specific-target genes we identified in other lineage-switching samples. As ENL accumulation was likely triggered by the oncoprotein, but ENL then remained bound to chromatin in the absence of the oncoprotein, this is an example of an epigenetic resistance mechanism that is unprecedented in the literature. Several therapeutics that target ENL have been developed and our results provide a strong justification for the pursuit of menin inhibitor/ENL inhibitor combination therapies for the treatment of lineage-switching and relapse refractory *KMT2Ar* leukemias.

The development and application of therapeutics that target chromatin-regulatory mechanisms is a red-hot field that is rapidly progressing. Novel methods that assess the efficacy of these targeted therapies directly in patient samples are urgently needed. This manuscript demonstrates that AutoCUT&RUN is the right tool for the job.

Reviewer #1, expertise in leukemia epigenetics (Remarks to the Author):

In this manuscript, Janssens et al. profiled the KMT2A-r (i.e., MLL-r) oncoprotein target genes in 34 samples (4 cell lines, 12 infant patients, and 18 Ped/Adult patients) leukemic blast using a state-of-the-art autoCUT&RUN method. The ratios between probing the KMT2A-N' vs C' signals allow enrichment of gene loci potentially bound by the KMT2A-r (only has the N'). The authors performed various clustering and principal component analyses and summarized an "oncoprotein score" that correlates with fusion partners (AFF1, ENL, AF9, etc.) and leukemia types (AML vs. B-ALL). They also utilized a B-ALL-to-AML lineage-switching sample to derive a panel of "non-canonical AML program" (42 genes). Finally, they observed the enrichment of "non-canonical AML program" in 2 MENIN-inhibitor resistant patients. Overall, the authors suggest the KMT2A-r oncoprotein binding targets can be dynamically (and epigenetically) shifted during leukemia progression and lineage-shift.

While the concept of this manuscript is highly interesting, there are several concerns

requiring further validation. The main issues are the overall correlative analysis and the n = 1 lineage-switching sample, which make the “conclusion” more like an “assumption”. Here are my specific comments:

We thank the reviewer for their constructive feedback. We have now addressed their concerns in full (see below). We profiled two additional lineage-switching samples. One of these samples lineage switched during treatment with the menin inhibitor, providing an unprecedented opportunity to investigate the changes in oncoprotein binding that accompany therapeutic resistance. We have also rewritten the discussion section to provide a more measured interpretation of our results and hope the reviewer will agree that our conclusions are now fully supported by the data.

1. “These differences in oncoprotein scores suggest that among leukemias bearing the most common *KMT2A* translocations the levels of the oncoprotein are extremely heterogeneous.” – Ideally, an RNA-seq or a Western blot of the *KMT2A*-r oncoprotein expression in these samples should be presented to cross-compare with the “oncoprotein score” defined in this study.

We found that western blots took upwards of 4 million cells to observe the oncoprotein, which is not practical for precious patient samples. Instead, we used qPCR to measure the oncogene expression levels. Specifically, we identified primer pairs that span the minimal *KMT2A*-fusion-partner exon junctions in a collection of the *KMT2A*::*AF4*, *KMT2A*::*ENL* and *KMT2A*::*AFDN*-rearranged samples we profiled by AutoCUT&RUN (see Extended Data Fig. 1c-e). We then used qPCR to compare the relative oncogene expression and the average oncoprotein scores between samples that share the same minimal *KMT2A*-fusion-partner exon junctions (see Fig. 1h).

“By performing oncogene specific qPCR on samples that share the same minimal *KMT2A*::*AF4*, *KMT2A*::*ENL* and *KMT2A*::*AFDN* exon junctions, we find the differences in oncoprotein scores we observed using CUT&RUN generally reflect differences in the expression levels of the oncogene (Fig. 1h, Extended Data Fig. 1c-e). Only two samples that both bear the *KMT2A*::*AFDN* rearrangement did not fit this trend (marked by a star and asterisk in Fig. 1h). The ML-2 cell line (star in Fig. 1h) lacks the wild-type copy of *KMT2A*, suggesting wild-type *KMT2A* may be required for efficient oncoprotein loading. The wild-type *KMT2A* C-terminal signal was detectable in the second sample (asterisk in Fig. 1h), but it is possible the oncoprotein loading efficiency is reduced in this sample through an alternative mechanism. We conclude the average oncoprotein score provides a semi-quantitative metric that is indicative of differences in the oncogene expression levels between samples. Among leukemias bearing the most common *KMT2A* translocations, the gene expression levels of the oncoprotein are highly heterogeneous.”

2. The authors selected GNAQ to present in fig. 1e. However, how GNAQ was selected was

not presented in the manuscript. A statistical analysis of 34 samples divided into Clusters 1, 2, 3, and 4 for their oncoprotein binding signal at GNAQ is necessary for evaluation. Similarly in Fig. 2, the selection of FLT3 and ZNF521 should include more statistical rationale derived from the dataset.

This was a very helpful suggestion. In light of the other reviewers comments we have now replaced GNAQ with MBNL1 in the Fig. 1, and added a similar analysis of MEIS1, JMJD1C and MEF2C to Extended Data Fig 2. These are the genes that are most frequently called as oncoprotein targets in our collection of samples and this is now made clear in the text. In addition we added a statistical comparison of the oncoprotein scores over the MBNL1, MEIS1, JMJD1C and MEF2C loci separated based on the 4 Clusters from the UMAP (see Fig. 1j and Extended Data Fig. 2b,e,f). This analysis revealed the MBNL1 and JMJD1C oncoprotein scores scale with the average oncoprotein scores in each of the four Clusters and that MEIS1 and MEF2C show a slightly different pattern. Specifically, MEIS1 oncoprotein scores are similar in Clusters 2, 3 and 4, and the MEF2C oncoprotein scores are lower in cluster 3 than 2.

3. Fig. 2f and 2g: The PC1 Gene Ontology (GO) (Fig. 2f) related to blood lineages is expected. However, the PC2 GO (Fig. 2g) related to other tissue development (skeletal, etc.) was not readily connected to these leukemia samples. Can the authors speculate as to what could be the relevance of these GOs to the examined KMT2A-r leukemias?

Excellent Point. Numerous genes in the HOXA cluster contribute to PC2 and were enriched in the GO terms the reviewer is referring to. To obtain an unbiased perspective of the genes in PC1 and PC2 we now add an analysis of the average expression pattern of the PC1 and PC2 genes in normal hematopoiesis and find PC1 genes are enriched in proB cell progenitors whereas PC2 genes are normally expressed in Hematopoietic Stem Cells and Multipotent Progenitors (see Fig. 2e-g). We hope the reviewer agrees this revised analysis helps to clarify the gene sets that are represented in PC1 versus PC2, and we thank them for raising this concern.

4. "The group of "highly sensitive" genes that a previous study found are rapidly downregulated in response to auxin-mediated KMT2A-AF9 degradation include genes that contribute to PC1 (MEF2C, CPEB2 and MEIS1) as well as PC2 (SKIDA1, PBX3, HOXA9 and SOCS2; Supplementary Information)" – The information listed in the "Supplementary Information" will benefit from a clearer description or data presentation. For example, the MEIS1 and SOCS2 genes each were found in both PC1 and PC2 gene lists in the "Supplementary Information".

We have now corrected this error: "The group of "highly sensitive" genes that a previous study found are rapidly downregulated in response to auxin-mediated KMT2A::AF9 degradation includes genes that contribute to both PC1 and PC2 (MEF2C, MEIS1 and

SOCS2) as well as PC1-specific genes (*CPEB2*) or PC2-specific genes (*SKIDA1, PBX3, HOXA9*; Supplementary Information)²².”

5. The current study derived a series of analyses based on 1 patient sample pairs that exhibited B-ALL-to-AML lineage-switching. The main concern is the reproducibility, as the conclusion was driven by $n = 1$. At least, the authors should speculate on additional factors that could bias the current conclusion, e.g., the “passive clonal selection” instead of “active switch” in this patient could also explain the lineage-switching. This also raises concerns as to whether the same switch-related signature could be observed in more patients experiencing lineage-switching during relapse.

We appreciate the reviewers understanding that for a variety of reasons that are beyond our control lineage-switching samples are challenging to obtain. However, during revisions we were able to add two additional lineage-switching samples. Because a post-switching sample was not available for one of them, and the other lineage-switched during treatment with the menin inhibitor, we were not able to directly measure the oncoprotein-binding sites in the AML after switching. To address the concern as to whether the switch-related signature we describe is a common feature of lineage switching, we (1) split out the data from the lineage-switching samples previously analyzed by RNA-seq in Tirtakusuma et al. 2022. The B-ALL-specific genes we identified were significantly downregulated in all 6 lineage-switching samples they profiled, and the AML-specific genes (now referred to as the GMP-like program) are upregulated in all 6 samples, but this difference was not significant in one sample. (2) We see ENL persists on the chromatin in the AML sample that lineage switched during treatment with the menin inhibitor, and the ENL signal is enriched over the GMP-like program. Together, we hope this revised analysis convinces the reviewer the GMP-like program has a recurring role during the lineage switching of *KMT2A*r leukemias.

In addition, we added a section to the discussion section to address the possibility that lineage switching may result from a trans-differentiation mechanism or “active switch” or alternatively could be the result of “passive clonal selection”:

“Lineage switching could result from the trans-differentiation of B-ALL cells to an AML identity, or from the selective outgrowth of a latent multipotent cell type that retains both B-ALL and AML potential. A strong shift in the oncoprotein levels might support a trans-differentiation event. However, multiple lines of evidence suggest a selective outgrowth model is more likely. First, our comparison of oncoprotein-binding sites did not readily distinguish the B-ALL lineage-switching samples from the other B-ALL samples we profiled. Second, whole-exome sequencing of samples before and after lineage switching revealed a paucity of shared mutations¹⁰. And third, studies using single-cell RNA-seq identified a rare population of myeloid-biased cells in lineage-switching samples¹⁸. The latent multipotent progenitor model predicts that while the *KMT2A::AF4* oncoprotein was thriving in the p279 B-ALL cells, the ALL-directed chemotherapy was effectively able to eliminate them. Then, the

p318 AML sample arose from the same latent population that gave rise to the p279 B-ALL cells.”

6. The enrichment of “non-canonical AML program” (42 genes) in 2 MENIN-inhibitor resistant patients is intriguing. As a comparison, an examination of the “non-canonical AML program” (42 genes) in MENIN-inhibitor sensitive patients should be presented in the manuscript to evaluate the specificity of this signature to MENIN-i resistance.

We thank the reviewer for this enthusiastic and constructive comment. In the revised manuscript we have added to this analysis the MV4;11 and PDX4 samples that showed a persistent response to the menin inhibitor (see Fig. 4j).

7. The authors concluded “We propose that a dynamic shift in the binding sites of the oncoprotein activates an alternate leukemia program and this epigenetic mechanism promotes resistance to targeted therapies.” – Indeed, there is no “epigenetic” measurement (e.g., DNA/histone modification, ATAC-seq, chromatin looping, etc.) presented in this manuscript. Furthermore, the potential contribution of “genetic” alterations (i.e., mutations) was not examined in this study (so, this possibility could not be excluded). I recommend the authors refine the data interpretation to consider alternative possibilities.

We now add profiling data of the oncoprotein cofactors menin, DOT1L and ENL in a sample that lineage switched and is resistant to the menin inhibitor. In this sample ENL persists on the chromatin, which is suggestive of a potential role of ENL in the epigenetic resistance to targeted therapy. That said, we agree with the author that we cannot currently disentangle the potential contributions of genetic and epigenetic mechanisms to the resistance of *KMT2Ar* leukemias to targeted therapies. We have revised our data interpretation and discussion accordingly:

“Intensive genetic and biochemical characterization has demonstrated menin, DOT1L and ENL act as transcriptional cofactors of the oncoprotein^{31,32,35-40}. Numerous compounds targeting each of these interactions have shown promise in preclinical models^{14,15,22,41-45}, and phase I clinical trials indicate menin inhibitors are effective for treating a subset of relapse refractory patients with *KMT2Ar* AML⁵. Several examples of genetic mechanisms that lead to lineage switching or acquired resistance to menin inhibitors have been described^{10,16,17}. Our results suggest *KMT2A* oncoproteins also promotes resistance to targeted therapies through epigenetic mechanisms. During lineage switching the oncoprotein activates a secondary GMP-like program, and ENL can persist on these genes even after the oncoprotein is effectively eliminated through treatment with a menin inhibitor. Although genetic alterations might have contributed to resistance in these samples, our results point

to ENL as a linchpin of epigenetic resistance to menin inhibitors and warrant investigation of combination therapies.”

Reviewer #2, expertise in KMT2A-rearranged leukemia (Remarks to the Author):

Janssens et al describe the analysis of a cohort of patient samples with KMT2A rearrangements that were analyzed with auto-Cut-and-Run for KMT2A-fusion binding. This is an extension of a previously published cohort interrogated with the same technique.

The key strength of this manuscript lies in the fairly large cohort and the use of primary patient samples. Key findings are the somewhat surprising inter-patient variability of binding sites, and the insights of how the fusion partner influences binding sites.

We thank the reviewer for their thorough critique of our work and their constructive feedback. We have addressed their comments in full.

This work has substantial limitations:

- Detection of the fusion is very indirect, which reduces the confidence of the identification. An IgG control is used to informatically remove over 2500 peaks, then C-terminal peaks are subtracted from N-terminal peaks to indirectly compute fusion binding. Sweeping statement made based on 40 residual peaks in a single sample.

We appreciate the reviewer’s attention to detail and are happy to have the opportunity to address their concerns. The 2500 peaks that are identified in the IgG control do not all overlap with the >100,000 peaks that are called in our combined KMT2A profiles, and those that do tend to be at highly repetitive regions in the pericentromeric and telomeric regions of the chromosomes. We felt the use of an IgG control to remove non-specific peaks was more appropriate than simply applying a repeat masker which is an alternative option.

- It is not possible to look at the quality of the primary data for the Cut&Run as the GSE given seems to have a typo – the data set that comes up relates to astrocyte inflammation.
- The GSE extension for the previously published samples is functional. Analysis of the primary data showed it to be very noisy, with not a good replicate concordance. The published detailed analysis in the prior publication suggests that it is possible to detect meaningful patterns on a cohort basis. However, the high degree of technical variability raises concern that that some of the variability that is the primary finding of this study could be technical rather than biological. Of particular concern are conclusions that rely on just one or two samples.

We apologize for this oversight. The reviewer token is now included so this reviewer can access all the beautiful new data. We agree that in a limited number of samples the data is sparse, but our analysis strongly suggests this is because the oncoproteins are expressed at much lower levels in these samples. Also, we would like to point out that in our current manuscript we worked with Dominik Otto to develop a novel computation tool that we feel improves the fidelity of oncoprotein-target gene calls and allows comparisons between samples in a way that is now fully supported by rigorous statistics. Also, we applied a stringent threshold for oncoprotein-peak calling and while it is possible we may have missed some genes, the likelihood that we include false positives is now very low. For example, when applied to the CD34+ sample that does not contain the *KMT2A* mutation this method calls only 3 false positives (see **Fig. 1c**).

- Given these concerns, more examples tracks of well described fusion targets should be shown, including the canonical well validated targets *HOXA*, *Meis1*, *MEF2C*. Representative examples should be shown in the main figure, with the cohort shown in the supplement to allow the reader to judge the quality and reproducibility of the Cut and Run without downloading the data.

We thank the reviewer for this helpful comment. More examples of genome browser tracks are now included throughout the manuscript. We include images of the *MEIS1* and *MEF2C* loci in Extended Data Fig. 2 and the *HOXA9* locus in Fig. 2.

- Several groups have reported a lower expression of *HOXA* cluster genes in *KMT2A*-*AFF1* rearranged leukemias compared to other samples (Trentin et al. 2009, Stam et al. 2010 and Kang et al. 2012). Instead, these leukemias seem to frequently express *IRX1*. Do the authors find consistent differences in the *HOXA* cluster based on the fusion partner? How about *IRX1*?

We thank the reviewer for this insightful suggestion. We now include a section addressing this point directly:

"The *HOXA9* locus also contributes to PC2, and the *HOXA9* oncoprotein scores scale with the overall oncoprotein levels in all the samples except those bearing *KMT2A::AF4* translocations (**Fig. 2d,o,p**, Supplementary Information). *HOXA9* was not called as an oncoprotein target in three out of six of the *KMT2A::AF4* samples from UMAP Cluster 4, which had the highest overall oncoprotein levels (Extended Data Fig. 3j). Previous studies identified a subset of infant *KMT2A::AF4*-rearranged samples that have an exceptionally high risk of relapse and are characterized by low expression of genes in the *HOXA* cluster, and high expression of the homeobox transcription factor *IRX1*^{8,26-28}. This gene expression pattern has also been reported in *KMT2A::AF4*-rearranged B-ALLs that went on to lineage switch¹⁰. We find that *IRX1* is a direct target of *KMT2A::AF4* in the three samples from Cluster 4 in which *HOXA9* is not called as an oncoprotein-target gene (Extended Data Fig.

3k-m, Supplementary Information). When measured by qPCR, we find the expression of *HOXA9* and *IRX1* are generally mutually exclusive (Fig. 2q). The p279 sample is a B-ALL that went on to lineage switch in response to ALL-directed chemotherapy and is the only sample that expresses both *HOXA9* and *IRX1*. In p318, the AML that resulted from the lineage switch of p279, we find *HOXA9* is downregulated whereas the expression of *IRX1* is maintained (Fig. 2q). Our data suggests there is a dichotomy in the direct regulation of *HOXA9* and *IRX1* by the KMT2A::AF4 oncoproteins which may contribute to lineage-switching events. Together, we conclude that at high levels of expression the oncoproteins activate genes that influence the lymphoid versus myeloid identity of the leukemia (Fig. 2r)."

- How do the results relate to ChIP studies on different fusions done in retroviral models? (MLLL-AF4 and using subtraction ChIP (N - C)

How do they relate to ChIP studies in patient samples? It would be nice to see Venn diagrams that show overlap of the targets identified here compared to prior studies and other models.

These results are not shown, but we compared our list of oncoprotein-target genes to those identified by ChIP-seq targeting the HA epitope tag in the KMT2A::AF9 oncoprotein transduced CD34+ cord blood cells from Olsen et al. 2022. While many of the classic oncoprotein target genes (e.g. MEIS1 and HOXA9) are bound in their samples and ours, we find that many of the direct targets they call, are not occupied by the oncoprotein or wild-type KMT2A in any of the samples we profiled. This suggests that using CD34+ transfected cells as a "ground truth" for real oncoprotein-target regions is not appropriate. Similar oncoprotein-target-gene lists are rarely made available, and we did not feel it would be appropriate to redo our analysis from Janssens et al. 2022 comparing our data to Guenther et al. 2008. Therefore, we profiled the oncoprotein-fusion partner AF4 as well as the cofactors DOT1L and ENL in the SEM cell line and three additional patient samples. We find DOT1L and ENL are both highly enriched over the regions we call as oncoprotein-target sites (see Fig. 11 and Extended Data Fig. 2j-l). We hope the reviewer will agree this internal validation provides sufficient evidence that our method for identifying oncoprotein-target genes is working as expected.

- The prior publication showed that Cut&Run peak profiles at fusion targets recapitulated the broader peaks first reported by Tom Milne's group. However, that analysis was done mostly on cell lines, and a small number patient samples. Does the broad versus narrow peak feature for N-only vs N+C peaks hold up in the larger patient sample cohort? Are there any differences based on fusion partner?

Yes, we see the oncoprotein shows the extended binding pattern from the promoter out into the gene bodies that has been described by us and others. This pattern is less pronounced in the KMT2A::SEPT6 fusion, but we described this result previously. However, we did not see any obvious differences between oncoproteins in the new samples we

profiled. We agree that a careful analysis of the oncoprotein distribution may yield further insights, but our current manuscript is focused on the interdependency of the oncoproteins with the lineage context in which they are expressed, and the role of dynamic oncoprotein binding in the epigenetic resistance to targeted therapies. We hope to revisit the suggested analysis in the future.

- There is a missed opportunity to conduct transcriptomic analysis in the same samples and integrate the two datasets. The authors use a published RNA-Seq experiment to interrogate transcriptional changes of specific loci – this cannot substitute for a careful and comprehensive characterization of fusion binding and matched gene expression.

We now provide comparisons to the relative gene expression as measured by qPCR throughout the manuscript (e.g. Fig. 1k, Extended Data Fig. 2 c,f,i).

- A correlation between fusion binding patterns and fusion partner was also shown in the prior publication.

Yes, but we hope the reviewer agrees that by increasing the number of primary patient samples we profiled from 8 to 32 we now provide a far more informed and thorough analysis of how the fusion partner influences oncoprotein-binding site selection.

- A single lineage switch sample is analyzed. It is very difficult to draw any firm conclusions from a single sample, particularly with a technique where the number of no-specific peaks exceeds the number of specific peaks by an order of magnitude, with raw data that is as noisy as what it uploaded in GSE 159608, and with the raw data for this sample unavailable. In order to draw any biological conclusions, this cohort should be expanded to include at least 4-5 additional samples. Finally, while auto-Cut&Run is suitable as a hypothesis generating technique for large numbers of samples, the more specific and precise ChIP is the more appropriate technique for low numbers of samples where greater weight falls on individual peaks being specific. Validation experiments in this extended (but still very small) lineage switch cohort should be performed by ChIP and integrated transcriptomic analysis of the same samples. While not likely available at a single institution, lineage switch is now common in an era of B-cell directed therapies, and it should be possible for the authors to put together a cohort of 5-6 samples total.

We are always interested in obtaining more primary *KMT2Ar* leukemia samples and we reached out to all the physician scientists that we are aware of with access to biobanks that potentially contain *KMT2Ar* lineage-switching samples. This search yielded two additional lineage-switching samples. However, because a post-switching sample was not available for one of them, and the other switched during treatment with the menin inhibitor, we were not able to directly measure the oncoprotein-binding sites in the AML after switching. To address the concern as to whether the switch-related signature we describe is a common

feature of lineage switching, we (1) split out the data from the lineage-switching samples previously analyzed by RNA-seq in Tirtakusuma et al. 2022. The B-ALL-specific genes we identified are significantly downregulated in all 6 lineage-switching samples they profiled, and the AML-specific genes (now referred to as the GMP-like program) are upregulated in all 6 samples, but this difference was not significant in one sample. (2) We see ENL persists on the chromatin in the AML sample that lineage switched during treatment with the menin inhibitor, and the ENL signal is enriched over the GMP-like program. Together, we hope this revised analysis convinces the reviewer the GMP-like program has a recurring role during the lineage switching of *KMT2A* leukemias.

- The authors then relate the lineage switch pattern to a revumenib resistant RNA-Seq analysis from another publication. How do the authors know that these samples weren't also lineage switch (or on the way there), and that, rather than the revumenib resistance, drove the gene expression profile? This part should be removed as the claims made by the authors are not supported by the experiment shown. In order to make such a claim, a much larger cohort, preferably pre and post revumenib, and linked Cut&Run and RNA-Seq profiling would need to be performed, ideally supported by functional validation what about this "non-canonical" AML program induces resistance. This endeavor seems outside the scope of a revision.

By incorporating and sample that lineage switched during treatment with the menin inhibitor, we have significantly revised this section of the manuscript.

- There is inconsistent use of nomenclature, using official gene symbols for some fusion partners (*KMT2A*, *AFF1*), but not others (*AF9*, *AF10*, *ENL*).

We thank the reviewer for pointing this out, we would certainly like to avoid any potential for confusion in the genes we are referring. For consistency, we changed the *AFF1* gene name to *AF4* throughout.

Reviewer #3, expertise in CUT&RUN and cancer (Remarks to the Author):

In this manuscript, the authors utilize Auto.CUT&RUN to examine *KMT2A* and *KMT2A*-oncoprotein localization in 34 *KMT2A* leukemia samples using Nterminal and Cterminal antibodies to differentially assess WT vs oncogene localization profiles. Using these data, the authors demonstrate heterogeneity in oncoprotein localization and observe switching of oncoprotein localization in the B-ALL to AML lineage switch sample included. With these data, the authors propose a model of oncoprotein shifting in *KMT2A* leukemias. I think the paper is very well written and clear, with exciting results that may be impactful to understanding molecular mechanisms in *KMT2A*-rearranged leukemias. I have the following points for clarification:

We thank this reviewer for their thoughtful critique. We have addressed the reviewers concerns in full and appreciate the additional clarity this added to our revised manuscript.

1) The authors detail using two N-terminal antibodies (recognize both the WT and oncoprotein) and two C-terminal antibodies in each of the 34 samples. From the methods description, I interpret the description to mean that the authors combine the AutoCUT&RUN data from each of antibodies, but I do not see any comparison of their specificity or signal. Do the individual N-terminal antibodies demonstrate similar patterns before combining the data and do the two C-terminal recognizing antibodies have similar patterns?

Yes, the signal from the two N and two C terminal antibodies is typically very concordant, and this was addressed in Janssens et al. 2022.

2) Related to the antibody choice and data presented, to calculate oncoprotein localization vs WT, the authors take N to C ratio. In Figure 1C, the authors demonstrate clear enrichment in the leukemia cells relative to the CD34+ WT cells. However, the browser tracks throughout the paper show clear N-terminal enrichment, but the C-terminal enrichment is very modest. Therefore, I wonder how robust the ratio calculation is; meaning that if the signal is very low, taking a ratio can be very skewed. While the example browser tracks throughout the manuscript have clear N-terminal enrichment and little C-terminal enrichment, is this true for all the ~1500 locations? Were there any examples where Cterminal was equivalent (or more) than N-terminal?

We thank the reviewer for this question and have added a genome browser track showing the USP5 promoter, which is called as a target of wild-type KMT2A in many of the samples we profiled (see **Fig. 1i**). In fact, these wild-type sites are defined by a similar or greater enrichment of the KMT2A C terminus as compared to the N terminus. The computational tool we developed to call oncoprotein-target sites incorporates the KMT2A N/C ratio as well as the p value of this difference to ensure that sites with very sparse N and C terminal signal are not called as false positives.

3) I realize that there are a lot of AutoCUT&RUN data presented in this manuscript, but the second half of the paper and some major conclusions rely on the single lineage switch sample that was profiled. While I appreciate that patient samples are difficult to obtain, and lineage switch samples are not common, having at least one other sample would really strengthen the conclusions drawn. For instance, in Figure 4A, are the non-overlapping AML peaks due to heterogeneity within lineage switch samples?

In the revised manuscript we added two additional lineage-switching samples. However, because a post-switching sample was not available for one of them, and the other switched during treatment with the menin inhibitor, we were not able to directly measure the

oncoprotein-binding sites in the AML after switching. To address the concern as to whether the switch-related signature we describe is a common feature of lineage switching, we (1) split out the data from the lineage samples previously analyzed by RNA-seq in Tirtakusuma et al. 2022. The B-ALL-specific genes we identified are significantly downregulated in all 6 lineage-switching samples they profiled, and the AML-specific genes (now referred to as the GMP-like program) are upregulated in all 6 samples, but this difference was not significant in one sample. (2) We see ENL persists on the chromatin in the AML sample that lineage switched during treatment with the menin inhibitor, and the ENL signal is enriched over the GMP-like program. Together, we hope this revised analysis convinces the reviewer the GMP-like program has a recurring role during the lineage switching of *KMT2A* leukemias.

4) It was never clear to me if levels of KMT2A-oncoprotein levels were ever examined in the study. The model (Fig 3G) shows that low KMT2A-oncoprotein tips to AML; so in the lineage switch sample is KMT2A-oncoprotein lower relative to B-ALL samples? And are protein levels of the oncoprotein lower in AML relative to B-ALL in general?

We found that western blots took upwards of 4 million cells to observe the oncoprotein which is not practical for precious patient samples. Instead, we used qPCR to measure the oncogene expression levels. Specifically, we identified primer pairs that span the minimal KMT2A-fusion partner exon junctions in a collection of the *KMT2A::AF4*, *KMT2A::ENL* and *KMT2A::AFDN* samples we profiled by AutoCUT&RUN (see Extended Data Fig. 1c-e). We then used qPCR to compare the relative oncogene expression and the average oncoprotein scores between samples that share the same minimal KMT2A-fusion-partner exon junctions (see Fig. 1h).

"By performing oncogene specific qPCR on samples that share the same minimal *KMT2A::AF4*, *KMT2A::ENL* and *KMT2A::AFDN* exon junctions, we find the differences in oncoprotein scores we observed using CUT&RUN generally reflect differences in the expression levels of the oncogene (Fig. 1h, Extended Data Fig. 1c-e). Only two samples that both bear the *KMT2A::AFDN* rearrangement did not fit this trend (marked by a star and asterisk in Fig. 1h). The ML-2 cell line (star in Fig. 1h) lacks the wild-type copy of *KMT2A*, suggesting wild-type KMT2A may be required for efficient oncoprotein loading. The wild-type KMT2A C-terminal signal was detectable in the second sample (asterisk in Fig. 1h), but it is possible the oncoprotein loading efficiency is reduced in this sample through an alternative mechanism. We conclude the average oncoprotein score provides a semi-quantitative metric that is indicative of differences in the oncogene expression levels between samples. Among leukemias bearing the most common *KMT2A* translocations, the gene expression levels of the oncoprotein are highly heterogeneous."

REVIEWERS' COMMENTS

Reviewer #2 (Remarks to the Author):

Janssens et al describe the analysis of a cohort of patient samples with KMT2A rearrangements that were analyzed with auto-Cut-and-Run for KMT2A-fusion binding. This is an extension of a previously published cohort interrogated with the same technique. The key strength of this manuscript lies in the fairly large cohort and the use of primary patient samples. Key findings are the somewhat surprising inter-patient variability of binding sites, and the insights of how the fusion partner influences binding sites.

The revised manuscript is substantially improved with respect to identification of fusion target loci. The AFF1 angle (as NOT reflecting fusion status) is very interesting and helpful. Confirmation of the HOXA/IRX1 expression patterns and link to fusion partner is nice to see. The newer data is now accessible on GEO, and the quality is improved compared to the earlier data – likely reflecting some fine tuning and increased experience of the team. This is also nice to see.

We thank the reviewer for these positive comments.

Unfortunately, the initially submitted work has substantial limitations, which are only partially addressed in the revision.

1. A key concern that remains are results rely on just one or two samples. This concern has not been addressed. In the initial manuscript, generalized statements were made based on a single switch sample. Based on reviewer feedback, the team included two additional samples, and indeed found the response to be quite heterogenous. These findings underscore the critical importance of including more than once sample into key experiments. Unfortunately the authors then go on to examine ENL patterns in a single Menin inhibitor resistant sample, and again make sweeping conclusions based on a this single sample. Ideally this would be removed from the work until a larger cohort can be put together. If the authors wish to retain these results, the conclusions have to be toned down substantially.
2. It should be mentioned that concern about conclusions based on a single sample were also voiced by reviewer 1.
3. The title “KMT2A oncoproteins induce epigenetic resistance to targeted therapies” makes no sense. If the results in the single Menin inhibitor resistant sample were to be taken at face value, then the fusion is actually gone from DNA. So something else maintains expression of genes required for transformation. The title should not refer to drug resistance since this is the not well substantiated part of the publication.
4. The presence of the elongation mark ENL on a subset of genes that retain expression speaks to the age-old conundrum of epigenetics – distinguishing the chicken from the egg. Does ENL reflect that these genes are being transcribed? Or is it driving transcription? The authors present no data to suggest it's the latter. In the absence of such data, it is much more likely that ENL reflects the GMP like state (well established to promote lineage switch) rather than drives it.

We appreciate the reviewer's attention to detail and helpful suggestions to improve the clarity and rigor of our manuscript.

In light of these issues, the following statements need to be very much toned down:

- a. Title: KMT2A oncoproteins induce epigenetic resistance to targeted therapies (relies on a simple sample, also makes no sense)

New Title: MLL oncoprotein levels influence leukemia lineage identities.

b. Abstract: "...can induce epigenetic lesions, marked by ENL, that support resistance to targeted therapies". Interesting theory but not substantiated by the data.

Replaced with the following sentence: "We propose MLL oncoproteins promote lineage-switching events through dynamic chromatin binding at lineage-specific target genes and may support resistance to menin inhibitors through similar changes in chromatin occupancy."

c. Discussion, first paragraph: the discussion of the GMP program during lineage switch needs to cite the work of Chen and colleagues that shows emergence of a GMP like cluster over the course of a lineage switch (Blood 2022).

We have added this sentence: "The emergence of GMP-like cells during lineage-switching was previously suggested by single-cell transcriptomic and accessibility profiling¹⁸, and our work demonstrates the transition to a GMP-like state is directly supported by the MLL oncoprotein."

d. Discussion, first paragraph: "We propose that KMT2A-oncoprotein dynamics and the induction of "epigenetic lesions," marked by ENL, play a critical role in the mechanisms that allow KMT2Ar leukemias to evade targeted therapies." This needs to be toned down given that it is relying on a single sample and no functional experiments support ENL as the "driver" of resistance.

Replaced with the following sentence: "We propose that MLL-oncoprotein dynamics support the activation of a GMP-like program as a secondary relapse mechanism and that ENL likely contributes to the maintenance of this program in *MLLr* leukemias that are resistant to treatment with menin inhibitors."

e. Discussion, last paragraph: "our results point to ENL as linchpin of epigenetic resistance to menin inhibitors". This conclusion is certainly not supported by correlative data in a single sample.

Replaced with this sentence: "Although genetic alterations might have contributed to resistance in these samples, our results indicate that ENL likely maintains the activation of the numerous MLL-oncoprotein target genes and warrant investigation of menin inhibitor and ENL inhibitor combination therapies."

5. This reviewer asked for a supplemental figure that shows key target loci (HoxA, Meis, Mef2c) for the C- and N-terminal MLL (both replicates), which would allow readers to easily get a sense of the quality of the data that underlies the conclusions without having to download and reanalyse the GEO file. This was not done.

The requested Figure is now provided as Supplementary Figure 1.

6. There is continued inconsistent use of nomenclature, using official gene symbols for some fusion partners (KMT2A, AFF1), but not others (AF9, AF10, ENL). The authors should pick one and stick with it (i.e. KMT2A-AFF1 / KMT2A-MLLT3 / KMT2A-MLLT10, or MLL-AF4 / ML-AF9 / MLL-AF10) etc.

The nomenclature has been modified for consistency throughout and we now use MLL::AF4 / ML::AF9 / MLL::AF10 etc.

Reviewer #3 (Remarks to the Author):

The authors have thoroughly addressed all the reviews from myself and the other reviewers. I think the addition of two more lineage switching lines has especially helped enhance the manuscript.

We thank this reviewer for their enthusiasm regarding our work, and their helpful feedback to strengthen our manuscript.